# Co-catabolism of arginine and succinate drives symbiotic nitrogen fixation

Carlos Eduardo Flores-Tinoco, Flavia Tschan, Tobias Fuhrer [ID], Céline Margot, Uwe Sauer [ID], Matthias Christen[*] [ID] & Beat Christen[**] [ID]

## Abstract

**Biological nitrogen fixation emerging from the symbiosis between bacteria and crop plants holds promise to increase the sustainability of agriculture. One of the biggest hurdles for the engineering of nitrogen-fixing organisms is an incomplete knowledge of metabolic interactions between microbe and plant. In contrast to the previously assumed supply of only succinate, we describe here the CATCH-N cycle as a novel metabolic pathway that co-catabolizes plant-provided arginine and succinate to drive the energy-demanding process of symbiotic nitrogen fixation in endosymbiotic rhizobia. Using systems biology, isotope labeling studies and transposon sequencing in conjunction with biochemical characterization, we uncovered highly redundant network components of the CATCH-N cycle including transaminases that interlink the co-catabolism of arginine and succinate. The CATCH-N cycle uses $N_2$ as an additional sink for reductant and therefore delivers up to 25% higher yields of nitrogen than classical arginine catabolism—two alanines and three ammonium ions are secreted for each input of arginine and succinate. We argue that the CATCH-N cycle has evolved as part of a synergistic interaction to sustain bacterial metabolism in the micro-oxic and highly acid environment of symbiosomes. Thus, the CATCH-N cycle entangles the metabolism of both partners to promote symbiosis. Our results provide a theoretical framework and metabolic blueprint for the rational design of plants and plant-associated organisms with new properties to improve nitrogen fixation.**

**Keywords** biological nitrogen fixation; *Bradyrhizobium diazoefficiens*; CATCH-N cycle; *Sinorhizobium meliloti*; TnSeq
**Subject Categories** Metabolism; Microbiology, Virology & Host Pathogen Interaction; Plant Biology
**Mol Syst Biol.** (2020) 16: e9419

## Introduction

Nitrogen is a fundamental element of all living organisms and the primary nutrient that impacts crop yield (Socolow, 1999). Despite being highly abundant in the atmosphere, plants can only assimilate nitrogen in reduced forms such as ammonium. More than 125 megatons of nitrogen are fixed annually by the industrial Haber–Bosch process into ammonium and applied to increase agricultural crop production (Graham & Vance, 2000). Endosymbiosis between legumes and soil bacteria termed rhizobia is capable to fix nitrogen biologically. On a global scale, anthropogenic nitrogen delivered to the environment surpasses annual supplies by natural biological nitrogen fixation on land (Gruber & Galloway, 2008) leading to serious environmental impacts from climate change to the disruption of eco-systems and pollution of coastal waters.

Improving the ability of plants and plant-associated organisms to fix atmospheric nitrogen has inspired biotechnology for decades (Beatty & Good, 2011; Bhardwaj *et al*, 2014; Gupta *et al*, 2015), not only for the apparent economic and ecological benefit that comes with the replacement of chemical fertilizers but also more recently for opportunities toward more sustainable agriculture and the potential to reduce greenhouse gas emissions. To catalyze atmospheric nitrogen fixation, rhizobia use a specific enzyme termed nitrogenase. Attempts to transfer and improve nitrogenase genes clusters have to date focused largely on organisms such as *Escherichia coli* (Dixon & Postgate, 1972; Wang *et al*, 2013). More recently, the emerging field of synthetic biology provides an alternative approach to engineer designer nitrogenase gene clusters in bacteria (Temme *et al*, 2012; Li *et al*, 2016; Burén *et al*, 2018; Yang *et al*, 2018). Despite these promising results, engineered organisms based on heterologous expression of nitrogenase genes have not yet come close to the efficiency of natural rhizobia–legume symbiosis systems (Beatty & Good, 2011; Good, 2018). While the molecular mechanism of the nitrogenase reaction has been resolved with atomistic detail (Hoffman *et al*, 2009, 2014; Seefeldt *et al*, 2009; Sippel & Einsle, 2017), the precise nature of metabolic interactions between plants and bacteria to sustain the energy-intensive process of nitrogen fixation has remained an open question.

The current model of nutrient exchange in rhizobia–legume symbiosis postulates that, in exchange for fixed nitrogen, the plant provides C4-dicarboxylic acids such as succinate, which is metabolized through the tri-carboxylic acid (TCA) cycle to generate ATP and reduction equivalents needed for the nitrogenase reaction (Watson *et al*, 1988; Yurgel & Kahn, 2004; Clarke *et al*, 2014). However,

Institute of Molecular Systems Biology, Eidgenössische Technische Hochschule (ETH) Zürich, Zürich, Switzerland
*Corresponding author. Tel: +41 44 633 76 58; Email: matthias.christen@imsb.biol.ethz.ch
**Corresponding author. Tel: +41 44 633 64 44; Email: beat.christen@imsb.biol.ethz.ch

multiple lines of evidence argue against a simple exchange of succinate for ammonium during symbiosis (Udvardi & Kahn, 1993; Kahn *et al*, 1985). The nitrogenase is highly sensitive to oxygen, which irreversibly inactivates the enzyme. While the microoxic environment encountered by rhizobia inside root nodules promotes nitrogenase activity, it also inhibits the catabolism of succinate through the TCA cycle. This is because increases in NADH and NADPH levels inhibit key enzymes of the TCA cycle including citrate synthase, isocitrate dehydrogenase, and 2-oxoglutarate dehydrogenase, a process termed redox inhibition (Dunn, 1998; Prell & Poole, 2006). Thus, the TCA cycle probably operates below its full aerobic potential. Furthermore, if the metabolism of symbiotic nitrogen-fixing bacteria is based exclusively on the provision of succinate, then the bacterial nitrogen requirement must be covered solely by the nitrogenase reaction. However, nitrogen-fixing root-nodule bacteria (termed bacteroids) do not self-assimilate but rather secrete large quantities of ammonium (Bergersen & Turner, 1967; Brown & Dilworth, 1975; Udvardi & Poole, 2013) suggesting that the plant provides the bacteroids with a nitrogen-containing nutrient to cover their nitrogen needs. Finally, the degradation product of succinate through a fully operational TCA cycle is carbon dioxide. However, it has been reported that nitrogen-fixing bacteroids also secrete the amino acids alanine and aspartate (Kretovich *et al*, 1986; Waters *et al*, 1998; Allaway *et al*, 2000) suggesting a partially operating TCA cycle to yield alanine or aspartate.

Based on metabolic considerations of inefficient TCA cycle operation under microaerobic conditions, the inability of bacteroids to self-assimilate nitrogen, and evidence for secretion of alanine or aspartate by nitrogen-fixing bacteroids, we postulated a nitrogen-containing nutrient that is plant-provided in addition to dicarboxylic acids. Since the plant must provide the N-containing compound in sufficient quantities, we reasoned that an amino acid might be a likely candidate. Based on the finding that nitrogen-fixing bacteroids utilize succinate and secrete the amino acids alanine and aspartate (Kretovich *et al*, 1986; Waters *et al*, 1998; Allaway *et al*, 2000; Day *et al*, 2001), we concluded that the plant-provided compound must comprise at least two nitrogen atoms to enable two consecutive transamination reactions. The first nitrogen is used for transamination of the ketoacid derived from succinate while the second nitrogen atom is utilized for transamination of the ketoacid derived from the plant-provided compound.

Six out of the twenty natural amino acids (Arg, His, Lys, Gln, Asn, and Trp) contain two or more nitrogen atoms and thus are likely candidates. Thereof, His, Lys, and Gln can be excluded because their degradation involves a compulsory 2-oxoglutarate dehydrogenase step, which is subjected to redox inhibition and disfavored under microoxic conditions (Salminen & Streeter, 1990, 1992). Furthermore, we also excluded Trp and Asn because their catabolism enters the TCA cycle at the level of pyruvate and oxaloacetate, respectively, which limits energy metabolism within a partially operating TCA cycle. Based on these theoretical considerations, we postulated that the remaining amino acid arginine is a likely candidate for the nitrogen-containing compound provided upon symbiosis.

Here, we report on the CATCH-N cycle based on the co-catabolism of plant-provided arginine and succinate as part of a specific metabolic network to sustain symbiotic nitrogen fixation as a synergistic interaction. Using $^{13}$C and $^{15}$N isotope tracing experiment in *Bradyrhizobium diazoefficiens* in conjunction with *in planta* transposon-sequencing analyses and enzymatic reaction network characterization in *Sinorhizobium meliloti*, we uncovered the principle of the metabolic inter-species interaction leading to the nitrogen-fixing symbiosis between plants and bacteria. Collectively, we demonstrate that the CATCH-N metabolism is governed by highly redundant functions comprised of at least 10 transporter systems and 23 enzymatic functions. In sum, our systems-level findings provide the theoretical framework and enzymatic blueprint for the optimization and redesign of improved symbiotic nitrogen-fixing organisms.

# Results

### The co-feeding of arginine and succinate stimulates nitrogenase activity

To probe whether arginine functions as co-substrate to drive symbiotic nitrogen fixation, we assayed nitrogenase activity of mature bacteroids from *B. diazoefficiens* (strain 110 spc4) and *S. meliloti* (strain CL 150) in the presence of nodule crude extracts and upon supplementation of succinate and arginine (Materials and Methods). The addition of nodule crude extracts to isolated bacteroids resulted in strong stimulation of nitrogenase activity (Fig 1A), supporting the idea that plant-provided nutrients are necessary for symbiotic nitrogen fixation. While the stimulation of nitrogenase only poorly occurred in the presence of succinate as the sole nutrient, we found that the addition of arginine stimulated nitrogenase activity in *B. diazoefficiens* and *S. meliloti* by 46% ± 4% and 116% ± 2%, respectively, as compared to nodule extracts (Appendix Table S1). The co-feeding of arginine in combination with succinate restored nitrogenase activity to the same extent as nodule extracts (91% ± 6% and 92% ± 6%) for *B. diazoefficiens* (Fig 1A) and *S. meliloti* bacteroids, respectively. Furthermore, adding solely malate or co-supplementing malate and arginine inhibited or only poorly stimulated the nitrogenase activity in isolated *B. diazoefficiens* bacteroids (−20% ± 5%, and 9% ± 4%, Fig 1A, Appendix Table S1). Therefore, we concluded that the co-feeding of arginine and succinate is sufficient to stimulate nitrogenase activity in bacteroids.

The nitrogenase enzyme complex catalyzes one of the most energy-consuming enzymatic reactions found in nature with 16 ATP molecules and 8 low-potential ("high-energy") electrons required for the reduction of a single nitrogen molecule. Nitrogenase is irreversibly inactivated in the presence of oxygen, which restricts the reduction of atmospheric nitrogen to low-oxygen conditions. Thus, to support nitrogen fixation, bacteroids must produce substantial amounts of ATP under microoxic conditions. The finding that succinate as the sole nutrient did not result in nitrogenase stimulation suggested that succinate catabolism via the TCA cycle does likely not generate sufficient ATP to support efficient nitrogenase reaction.

To measure the ATP level produced in isolated *B. diazoefficiens* bacteroids, we quantified the increase in intracellular ATP through ATP-dependent luciferase assays (Materials and Methods). In agreement with the absence of an operational TCA cycle, we observed that the addition of succinate alone failed to stimulate ATP production. In contrast, we found that co-feeding of succinate together with arginine caused an increase from 1.53 ± 0.04 to 4.03 ± 0.09

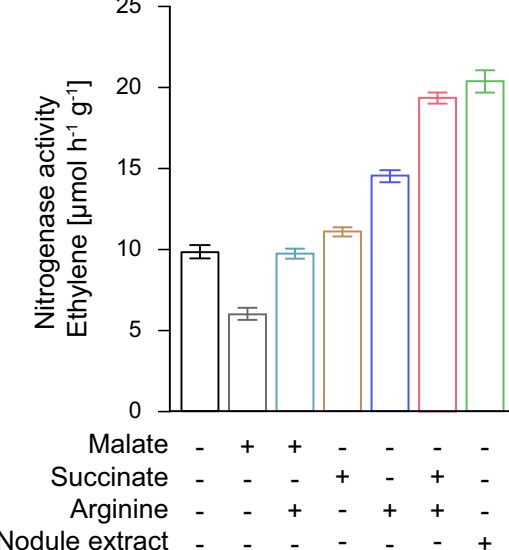

**A** Nitrogen fixation in bacteroids

| | | | | | | |
|Malate|−|+|+|−|−|−|−|
|Succinate|−|−|−|+|−|+|−|
|Arginine|−|−|+|−|+|+|−|
|Nodule extract|−|−|−|−|−|−|+|

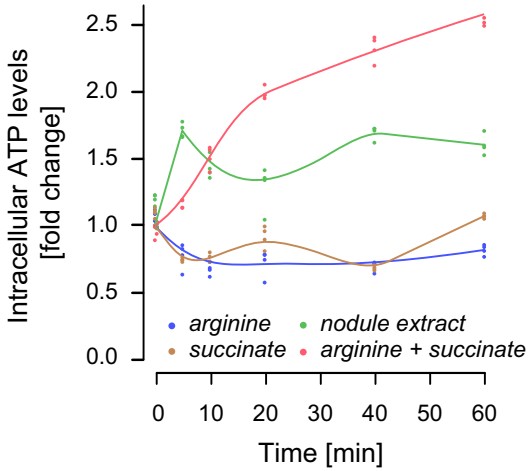

**B** ATP production in bacteroids

- arginine
- succinate
- nodule extract
- arginine + succinate

---

**Figure 1. The co-feeding of arginine and succinate promotes nitrogen fixation and ATP production in isolated bacteroids.**

A Substrate-dependent nitrogenase activity, measured through reduction of acetylene into ethylene, in isolated *B. diazoefficiens* bacteroids upon supplementation of malate, succinate, arginine, and nodule extract. Data represent the mean and standard error of the mean of at least eight independent replicates.

B Fold change in intracellular ATP level in isolated *B. diazoefficiens* bacteroids upon supplementation of succinate, arginine, and nodule extract. The values obtained for each of the four individual measurements (points) together with the regression trajectory of each experiment are shown.

Source data are available online for this figure.

---

attomole ATP per cell corresponding to a $2.61 \pm 0.03$-fold increase in intracellular ATP levels (Fig 1B). In sum, these findings demonstrate that co-catabolism of arginine and succinate supports biological nitrogen fixation *in vitro* in *B. diazoefficiens* and *S. meliloti*.

## Isotope tracing experiments reveal the presence of three parallel arginine degradation pathways

To gain further insights into the bacteroid metabolism and possible routes of arginine degradation operating during nitrogen fixation, we performed stable isotope labeling studies with *B. diazoefficiens*. We incubated isolated bacteroids under stringent microoxic conditions with $^{13}$C arginine in the presence of unlabeled succinate and quantified isotope labeling pattern of arginine degradation intermediates by LC-MS/MS (Table 1). Upon the addition of $^{13}$C arginine, we observed a rapid increase in the labeled intracellular arginine pool (99.43% $^{13}$C), demonstrating active arginine transport into nitrogen-fixing bacteroids.

Upon further incubation, we found $^{13}$C isotope labels in key intermediates of multiple arginine degradation pathways steadily increasing. After 150 min, ornithine, proline, and glutamate, the intermediates of the classical arginase-mediated degradation pathway, were labeled to 81.25%, 23.19%, and 7.67% $^{13}$C (Table 1). In addition, we found that citrulline, which represents the first step of the arginine deiminase pathway, was labeled to 60.22% $^{13}$C. The presence of an arginine deiminase pathway in isolated bacteroids is in agreement with the previously proposed enzymatic production of ATP by the enzyme carbamate kinase (Dunn, 2015).

Furthermore, we also observed fractional labeling of 90.40% for 4-guanidinobutanoate and 6.26% for 4-aminobutanoate (GABA), suggesting the presence of a functional arginine transaminase pathway operating in bacteroids (Table 1) that yields alanine or aspartate. The observation that bacteroids possess an arginine transamination pathway was intriguing because it provides a functional link between arginine degradation and alanine or aspartate secretion, which was previously reported as part of the metabolite exchange occurring during symbiotic nitrogen fixation (Day *et al*, 2001). In agreement with this hypothesis, upon incubation of isolated bacteroids under stringent microoxic conditions with $^{15}$N arginine, we observed fractional labeling of aspartate of 90.34% (Appendix Fig S1). These findings suggest that at least three independent arginine degradation pathways operate simultaneously in nitrogen-fixing *B. diazoefficiens* bacteroids causing release of ammonium independent from the nitrogenase reaction.

## Transposon sequencing reveals symbiosis genes involved in the uptake and catabolism of arginine

To gain further insights into the gene sets and enzymatic functions responsible for uptake and degradation of arginine, we conducted a

**Table 1. Arginine catabolism in *Bradyrhizobium diazoefficiens* bacteroids fed with $^{13}$C arginine and unlabeled succinate.**

| Metabolite | Fractional labeling (%)[a] |
|---|---|
| Arginine (ARG) | 99.43 ± 0.10 |
| Citrulline (CIT) | 60.22 ± 5.26 |
| Ornithine (ORN) | 81.25 ± 5.01 |
| Proline (PRO) | 23.19 ± 3.58 |
| Glutamate (GLU) | 7.67 ± 0.82 |
| 4-guanidinobutanoate (GBA) | 90.40 ± 3.03 |
| 4-aminobutanoate (GABA) | 6.26 ± 1.06 |

[a]$^{13}$C Fractional labeling after 150 min incubation with $^{13}$C L- arginine. Shown are the average and the standard error of the mean (SEM).

functional genetic screen *in planta* using transposon sequencing (TnSeq; van Opijnen *et al*, 2009; Christen *et al*, 2011). TnSeq measures genome-wide changes in transposon insertion abundance prior and after subjecting large mutant populations to selection regimes (Christen *et al*, 2016) and allows systems-level definition of conditional essential gene sets for a given environment (Ochsner *et al*, 2017; Québatte *et al*, 2017). We reasoned that TnSeq provides a unique opportunity to identify specific metabolic pathways including arginine transport and degradation genes that become essential upon engagement in symbiosis.

We choose *S. meliloti-Medicago truncatula* as the rhizobia–legume symbiosis system, because supernodulating *M. truncatula* lss plants (Schnabel *et al*, 2010) provided a high frequency of nodules increasing the resolution of the TnSeq analysis. In total, we infected 4,500 *M. truncatula* lss plants with a high-density *S. meliloti* transposon mutant library of 750,128 unique Tn5 insertions (Fig 2A, Dataset EV1, Materials and Methods). Six weeks post-inoculation, we recovered 99,623 unique Tn5 mutants from 375,000 root nodules (Dataset EV2). By comparing the TnSeq dataset obtained from *in planta* infection assays and input transposon mutant libraries, we mapped a set of 977 symbiosis genes corresponding to 15.71% of the tripartite 6.7-megabase (Mb) genome (Dataset EV3, Materials and Methods). A gene is classed as a symbiosis gene when the fractional representation in the library recovered from nodules is significantly less that the fractional representation in the original library (Dataset EV3, Materials and Methods), implying that strains carrying the mutation do not prosper in nodules. Among the identified 977 symbiosis genes, 435 genes were located on the chromosome, 295 on pSymA, and 247 on pSymB indicating that all three replicons of *S. meliloti* contribute to symbiotic nitrogen fixation (Fig 2B).

Functional classification revealed that the large majority of symbiosis genes comprise cellular functions such as metabolism (507 genes, 51.89%), gene regulation (196 genes, 20.06%), and other cellular processes (228 genes, 23.34%; Fig 2C, Dataset EV4, Materials and Methods). The identified gene set included well-characterized symbiosis factors involved in nodulation (34 genes, 3.48%) as well as functions associated with the nitrogenase enzyme complex (12 genes, 1.23%). Collectively, a set of 177 symbiosis genes, corresponding to over one-third of the 507 metabolic symbiosis genes (34.91%), was associated with nitrogen metabolism including genes for the transport (59 genes), biosynthesis (78 genes), and degradation (40 genes) of amino acids and other nitrogen-containing compounds. While only 3 out of 78 essential biosynthesis genes (3.85%) were involved in the synthesis of arginine, we found a large fraction of 18 out of 59 essential transport genes (30.51%) and 22 out of 40 essential catabolic genes (55.00%) annotated as being involved in the uptake and catabolism of arginine and its derivatives. In sum, these findings from *in planta* TnSeq analysis highlight that the provision of arginine and its consecutive degradation is of fundamental importance to drive symbiotic nitrogen fixation *in planta*.

## TnSeq identifies multiple arginine transport systems mediating acid tolerance

Among the 18 transport genes essential for symbiosis, we found two putative arginine and four putrescine ABC transport systems that we named *artABCDE* (SMc03124-28) for arginine transporter; *satABC* (SMa2195-97) for symbiotic arginine transporter; and *potFGHI*

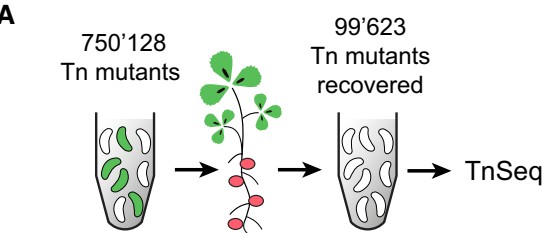

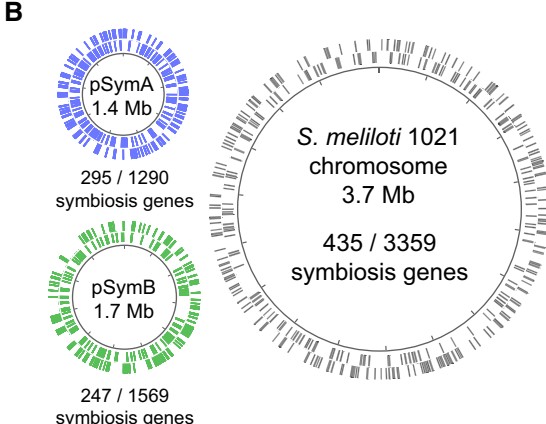

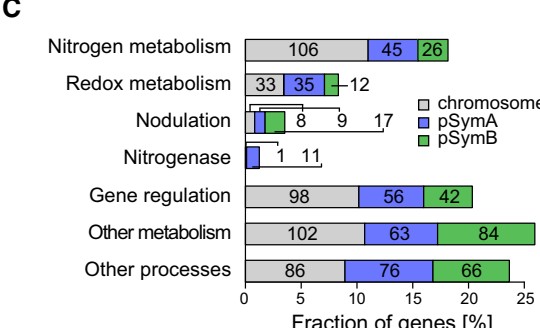

**Figure 2. The symbiosis genome of *S. meliloti* revealed by transposon sequencing (TnSeq).**

A  Schematic representation of the plant infection screen that was used to map the *S. meliloti* symbiosis genome. Tn5 transposon mutant pools were selected for their ability to establish symbiosis with *M. truncatula*. After selection, Tn5 mutants recovered from root nodules were identified by TnSeq.

B  Genome map visualizing the distribution of essential symbiosis genes among the three *S. meliloti* replicons. Symbiosis genes are plotted as lines on the chromosome (gray) and the mega-plasmids pSymA (blue) and pSymB (green).

C  Functional classification of essential symbiosis genes located on the chromosome (gray), pSymA (blue), and pSymB (green).

Source data are available online for this figure.

(SMc00770-3), *potABCD2* (SMa0799-803), *potABCD3* (SMa0951-3), and *potABCD4* (SMa2203-9) for the putrescine uptake systems (Dataset EV3). In addition, two arginine/agmatine antiporter genes *adiC* (SMa0684) and *adiC2* (SMa1668) encoded on pSymA were also essential during symbiosis (Dataset EV3). Interestingly, the identified transport systems participate in urease reactions and arginine deiminase pathways that mediate acid tolerance. Cross-comparing

expression profiles using previously published RNA-seq datasets (Roux *et al*, 2014), we found that *artABCDE* was the only transport system to be constitutively expressed during all stages of symbiosis, while the expression of all other transporters was specifically induced during development into nitrogen-fixing compartments (symbiosomes).

To gain further insights, we searched for additional symbiosis genes related to acid tolerance and indeed found multiple essential components in the TnSeq dataset (Dataset EV3). From the urease pathway, we identified two arginase genes *argI1* (SMc03091) and *argI2* (SMa1711) and the urease gene *ureA* (SMc01941) and *ureE* (SMc01832). From the arginine deiminase system, we found the *arcABC* operon (SMa0693, SMa0695, SMa0697) to be essential for symbiosis. Both systems catalyze the conversion of arginine into ornithine leading to the production of ammonia as part of the acid tolerance mechanism. Furthermore, the arginine deiminase system also provides ATP via the enzymatic step of ornithine carbamoyl-transferase *arcB* (Cunin *et al*, 1986). Interestingly, two additional copies of ornithine carbamoyltransferase were also essential (*argF1*, encoded by SMc02137, and *arcB2*, encoded by SM_b20472), emphasizing the importance of genetic redundancy in arginine deiminase-dependent ATP synthesis during symbiosis. The urease and arginine deiminase acid tolerance mechanisms rely on the efflux of ammonium (Marquis *et al*, 1987). Indeed, the ammonium efflux pump encoded by *amtB* (SMc03807) was among the top-ranked symbiosis genes. These findings underscore the importance of ammonium secretion as a compulsive property of bacteroids independent of the nitrogenase reaction.

## Arginase gene deletions show nitrogen starvation phenotypes during plant infection assays

To validate the importance of the identified arginine-dependent acid tolerance systems for symbiosis, we constructed a panel of deletion mutants of the urease and arginine deiminase pathways and assessed nitrogen starvation phenotypes during plant infection assays (Materials and Methods, Fig 3D, Appendix Table S2). Out of the 8 mutants evaluated, all displayed symbiotic defects. On the level of the arginine transport systems, we found that *artABCDE* and *satABC* showed a reduction in nitrogenase activity of $47.06\% \pm 7.27\%$ and $55.45\% \pm 10.99\%$. Similarly, gene deletions in the urease pathway such as the arginase mutants *argI1* and *argI2* exhibited a reduction of $71.18\% \pm 5.21\%$ and $70.97 \pm 5.16\%$ for single deletions and $80.89\% \pm 3.15\%$ for the double deletion mutant. Deletion of the urease *ureGFE* and the ammonium efflux system *amtB* resulted in a $64.68\% \pm 5.50\%$ and $80.90\% \pm 4.81\%$ reduction in nitrogenase activity.

Plants inoculated with the *argI1*, *argI2* single and double deletion mutant harbored a typical phenotype of nitrogen starvation. The aerial part of infected plants was smaller than those inoculated with WT strain (Figs 3A and EV1). Nodules induced by the *argI1*, *argI2* double deletion mutant displayed the yellowish color of non-functional *M. truncatula* nodules (Fig 3B). Furthermore, observations of *argI1*, *argI2* nodule sections by scanning electron microscopy showed that nodules were hollow and remaining bacteroids exhibit aberrant cell morphology (Fig 3C, Appendix Fig S2). Collectively, our results suggest that the identified arginine-dependent acid tolerance system is a prerequisite for the faithful establishment of symbiosis.

## Identification of AspC as an arginine:pyruvate transaminase

In our isotope labeling studies with bacteroids, we detected fractional labeling of 90.40% for 4-guanidinobutanoate (Table 1) suggesting the presence of a functional arginine transamination pathway. However, in the *S. meliloti* genome, corresponding genes have not been assigned. Among the essential symbiosis genes identified by our TnSeq analysis (Dataset EV3), we found *aspC* (SMc02262) annotated as an aspartate transaminase, which is specifically expressed in nitrogen-fixing bacteroids (Roux *et al*, 2014). We constructed a clean deletion mutant of *aspC* and found that the absence of AspC indeed causes a nitrogen starvation phenotype during plant infection assays (Materials and Methods). The aerial part of plants infected with the *aspC* deletion mutant were smaller than those inoculated with wild-type strain, and nitrogenase activity was reduced by $50.91\% \pm 9.00\%$ (Fig 3D, Appendix Table S2) highlighting the importance of *aspC* for nitrogen fixation.

AspC shares 40% sequence homology to AruH, the arginine:pyruvate transaminase from *Pseudomonas aeruginosa* (Yang & Lu, 2007a, b). When we characterized the enzymatic properties of AspC by mass spectrometry, we found transaminase activity to pyruvate from arginine, agmatine, ornithine, and putrescine (Appendix Table S3) to yield alanine. Enzyme studies with *ilvB1* (SMc02263), the gene encoded downstream of *aspC*, further provided evidence that the *S. meliloti* arginine transamination pathway differs from *P. aeruginosa*. Sequence homology with AruI from *P. aeruginosa* suggested that IlvB1 is a 5-guanidino-2-oxopentanoate decarboxylase. However, assays with purified IlvB1 revealed that the *S. meliloti* enzyme has sevenfold higher decarboxylase activity with 5-amino-oxopentanoate. We concluded that both AspC and IlvB1 are promiscuous enzymes accepting multiple substrates. Although *aspC* was among the symbiosis genes, the deletion mutant retained partial nitrogenase activity. Also, *ilvB*1 was not present within the group of symbiosis genes. This suggests that genetic redundancies among transaminase and decarboxylase genes exist and partially mask mutant phenotypes complicating the genetic identification of additional network components.

## At least 16 redundant enzymes participate in the arginine transamination network

To further dissect possible redundant components, we performed a functional homology search for enzyme candidates known to be expressed during the nitrogen-fixing bacteroid stage. Conversion from arginine into succinate proceeds by a series of six consecutive enzymatic reaction steps. The four steps upstream of GABA comprise transamination, decarboxylation, ureohydrolase, and dehydrogenase reactions. In the lower part of the reaction network, a linear pathway through transamination and subsequent dehydrogenase steps leads from GABA to succinate (Fig 4A).

Upon heterologous expression and protein purification, we determined substrate specificity and biochemically profiled a panel of 16 candidate enzymes. In addition to the arginine deiminases ArcA1 and ArcA2 and the arginase ArgI1, we found two agmatinase ArgI2 and SpeB, and one ureohydrolase SpeB2 acting on 4-guanidinobu-tyraldehyde, 5-guanidino-2-oxopentanoate, and 4-guanidinobu-tanoate (Fig 4B, Appendix Table S4). The highest level of pathway redundancy resides on the level of dehydrogenases. Besides the five known GabD1-5 proteins from *S. meliloti*, we identified four

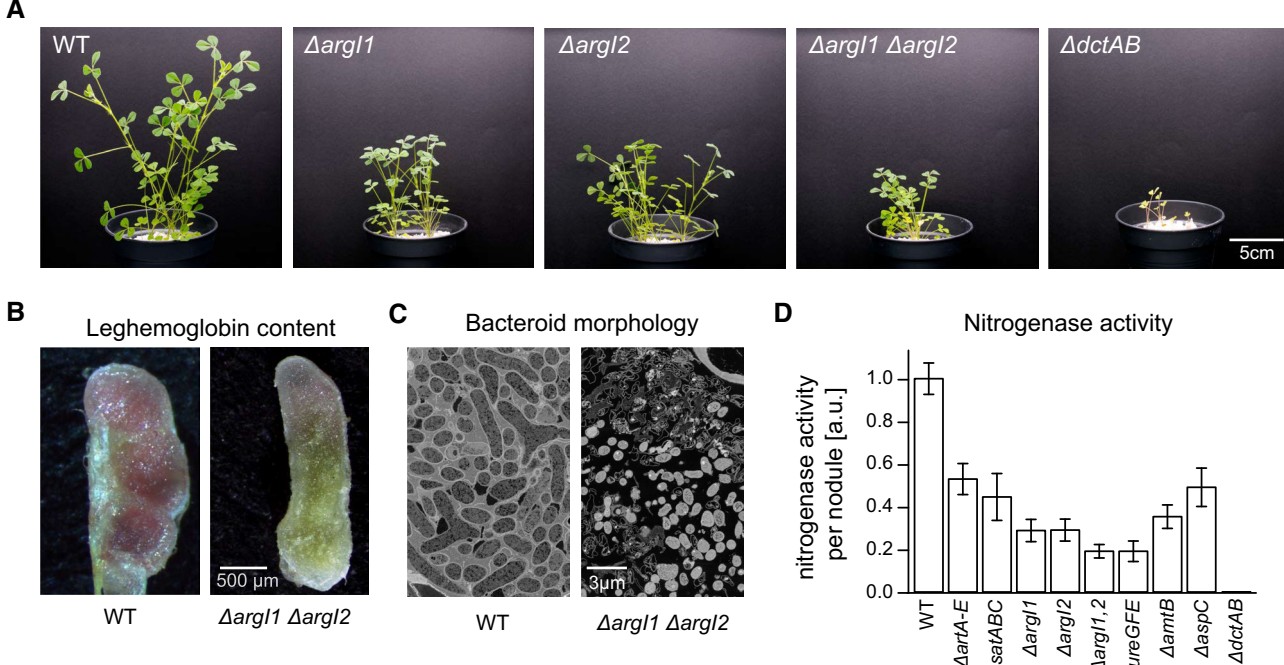

**Figure 3.  Assessment of nitrogen starvation phenotypes of *M. truncatula* upon infection with *S. meliloti* mutants impaired in arginine transport and catabolism.**

A   The aerial part of *M. truncatula* upon infection with *S. meliloti* Δ*argI1*, Δ*argI2*, Δ*argI*1,2 and the dicarboxylate transport mutant Δ*dctAB* were smaller than those inoculated with the wild-type strain, highlighting the importance of arginine catabolism for nitrogen fixation.

B   Cross sections of nodules bearing *S. meliloti* WT or arginine catabolism mutant Δ*argI1*, Δ*argI2* reveals the yellowish color of non-functional nodules induced by the Δ*argI1* Δ*argI2* double deletion mutant.

C   Bacteroid ultrastructure across nodule sections determined by scanning electron microscopy indicates the presence of hollow nodules with aberrant cell morphology in the Δ*argI1* Δ*argI2* double deletion mutant defective in arginine catabolism.

D   Nitrogenase activity in *M. truncatula* nodules inoculated with *S. meliloti* strains defective in arginine catabolism. Data points are the mean of at least 30 plants measured after 8 weeks post-inoculation; error bars indicate standard error of the mean.

Source data are available online for this figure.

additional isoenzymes Gab6-9 (Appendix Table S5). Thereof GabD6 and GabD7 share a dehydrogenase profile identical to GabD1 for 4-aminobutyraldehyde, succinic semialdehyde, and 4-guanidinobutyraldehyde while GabD8 and GabD9 exhibited substrate specificity for 4-guanidinobutyraldehyde (Fig 4B, Appendix Table S5).

Furthermore, on the level of pyruvate transaminases, we identified three additional enzymes DatA, AatB, and ArgD that exhibited substrate preferences for ornithine, putrescine, and agmatine and two enzymes GabT2 and GabT3 with preferences for GABA (Fig 4B, Appendix Table S6). Similarly, we also profiled two additional decarboxylase enzymes OdcA and OdcB that either catalyzed the decarboxylation of ornithine or 5-amino-oxopentanoate (Fig 4B, Appendix Table S7). Collectively, these results demonstrate the presence of a highly redundant network mediating arginine catabolism in *S. meliloti*.

**Synthetic reconstitution of the arginine transamination network that operates in nitrogen-fixing bacteroids**

We reconstituted the reaction system *in vitro* from a set of 14 enzymes and followed the conversion of arginine and pyruvate by mass spectrometry (Fig 4C, Appendix Table S8, Materials and Methods). We observed that 90% of the arginine was rapidly metabolized within

30 min. As expected, ornithine and 5-guanidino-2-oxopentanoate appeared as the first intermediates and then concomitantly decreased with the appearance of the second level of intermediates putrescine, 5-amino-oxopentanoate, and 4-aminobutyraldehyde that ultimately converted into GABA, succinic semialdehyde, and succinate. During the process, alanine steadily increased demonstrating that pyruvate transamination couples the conversion of arginine into succinate (Fig 4D, Appendix Table S8). In sum, these findings demonstrate the synthetic reconstruction of the transamination network that permits the co-catabolism of arginine and succinate.

**The catabolism of succinate and arginine is interlinked**

Since the arginine transamination network consumes two equivalents of pyruvate but generates only a single equivalent during its operation, we reasoned that the network strictly depends on the provision of additional pyruvate, which must be formed by simultaneous co-catabolism of succinate. We concluded that the degradation of arginine and succinate is mutually coupled and can only take place if plants provide both nutrients in equal stoichiometries. Indeed, single deletion in the succinate transporter DctAB abrogates succinate uptake and thereby also prevents the co-catabolism of

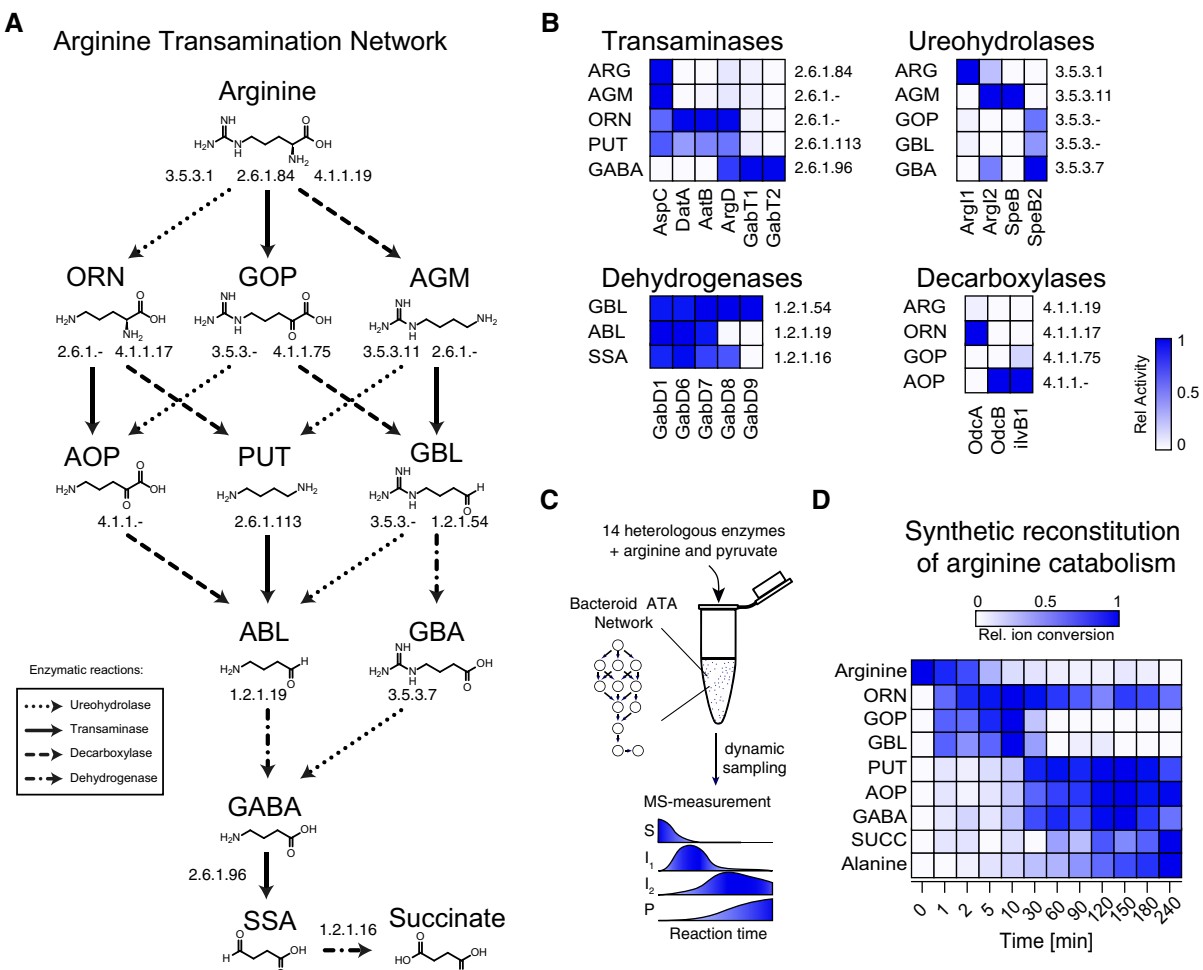

**Figure 4. Reaction scheme, enzyme profiling, and reconstitution of the catabolic arginine transamination network operating in nitrogen-fixing bacteroids.**

A Schematic representation of the arginine transamination reaction network that leads from arginine to succinate. Enzymatic reaction steps are indicated by arrows and annotated with corresponding enzyme commission (EC) numbers. Leftward facing dotted arrows represent ureohydrolase, rightward facing dashed arrows decarboxylases, downward-pointing solid arrows aminotransferases, and dashed and dotted arrows dehydrogenase enzyme reaction steps. The names of the metabolites are indicated above their chemical structures according to the following abbreviations: arginine (ARG), ornithine (ORN), 5-guanidino-2-oxopentanoate (GOP), agmatine (AGM), 5-amino-oxopentanoate (AOP), putrescine (PUT), 4-guanidinobutyraldehyde (GBL), 4-aminobutyraldehyde (ABL), 4-guanidinobutanoate (GBA), 4-aminobutanoate (GABA), succinic semialdehyde (SSA), and succinate (SUCC).

B Heat map representing enzymatic activities participating in the arginine transamination network with relative enzyme activities highlighted as a color map (Appendix Tables S4–S7). Rows are annotated on the left with the abbreviated names of the substrates and on the right with the corresponding EC numbers. Columns are annotated by corresponding enzyme names.

C Recombinant enzymes involved in arginine catabolism were combined into a single reaction mixture using arginine and pyruvate as substrates. Substrate (S) consumption into the products (P) succinate and alanine, including their intermediates ($I_1$, $I_2$), was determined along the time series using LC-MS/MS.

D The relative metabolite abundance along the reaction time-course is highlighted as a color map (Appendix Table S8).

Source data are available online for this figure.

arginine, resulting in a fix minus phenotype. On the other hand, the uptake of arginine is controlled by multiple redundant transporters. Nevertheless, single deletions in *artABCDE* and *satAB* arginine transporter show a partial symbiosis phenotype. Thus, transamination enforces a strict co-catabolism of succinate and arginine but also provides an elegant solution to maintain a partial TCA cycle under stringent microoxic and acidic conditions. We termed the entangled catabolic network CATCH-N cycle for C4-dicarboxylate arginine transamination co-catabolism under acidic (H⁺) conditions to fix nitrogen (Fig 5).

## A bifurcated electron transport chain operates during nitrogen fixation

We reasoned that the operation of the CATCH-N cycle provides significant amounts of NADH as well as $QH_2$. Under aerobic conditions, NADH is regenerated by the electron transport chain, which includes proton-pumping enzymes known as complexes I, III, and IV. However, upon symbiosome acidification, the driving force of complex I is likely no longer sufficient to sustain proton translocation against the increased pH gradient impairing the conversion of

NADH into $QH_2$. Also, oxygen partial pressure within symbiosomes is too low to operate the aerobic version of complex IV and bacteroids induce expression of a high-affinity cytochrome cbb3 oxidase complex FixNOQP1-3 (Preisig et al, 1993; Nellen-Anthamatten et al, 1998; Buschmann et al, 2010). Recently, the electron bifurcating FixABCX protein complex has been proposed to serve as the alternative entry point for low-potential ("high-energy") electrons from NADH (Ledbetter et al, 2017; Buckel & Thauer, 2018), thereby bypassing the impaired complex I. Electron bifurcation splits an electron pair into a endergonic and an exergonic redox reaction to yield a high and a low-potential electron (Müller et al, 2018).

Based on these findings, we devised a model to restore electron flow from NADH to $QH_2$ by electron bifurcation to nitrogenase and the high-affinity terminal oxidase (Fig 5). If this is the main pathway that permits regeneration of NADH, then all components must be essential in symbiosis. Indeed, we found genes encoding for components of the nitrogenase nifHDK (SMa0825, SMa0827, SMa0829), the electron bifurcation complex fixABCX (SMa0816, SMa0817, SMa0819, SMa0822), and the alternative complex IV fixNOQP1-3 among the top-ranked symbiosis genes in the TnSeq dataset (Dataset EV3). These genetic evidences support a model in which the CATCH-N cycle is interlinked with a bifurcated electron transport chain to permit nitrogen-fixing symbiosis.

## Estimation of the ATP balance of the bifurcated electron transport chain

The endergonic branch of the electron bifurcation reaction generates low-potential reducing equivalents in the form of flavodoxin hydroquinone ($Fld^{hq}$) for nitrogenase catalysis (Fig 5). For every $Fld^{hq}$, the nitrogenase consumes two additional ATP molecules. However, the exergonic branch of the electron bifurcation reaction translocates only three protons corresponding to a single ATP that is generated per electron passing from $QH_2$ to coenzyme Q onto oxygen. Thus, the electron bifurcation of each NADH appears to be associated with a net loss of one ATP. In addition to NADH, the CATCH-N cycle also provides $QH_2$ via succinate dehydrogenase (Fig 5). Thereby, up to two ATP are generated per $QH_2$ passing its electrons onto oxygen. Accordingly, a bifurcated electron transport chain in combination with an active succinate dehydrogenase complex delivers a net gain in ATP, provided that catabolism generates NADH to $QH_2$ in a 2:1 ratio. Contrary to this criterion, the complete TCA operating with malate or succinate provides a higher ratio of NADH to $QH_2$ of 5:1 and 5:2 and, thus, inevitably results in a net loss of ATP.

In addition to proton-motive force-dependent ATP synthesis, several metabolic cycles including the CATCH-N and TCA provide additional ATP through enzyme-coupled synthesis. Furthermore,

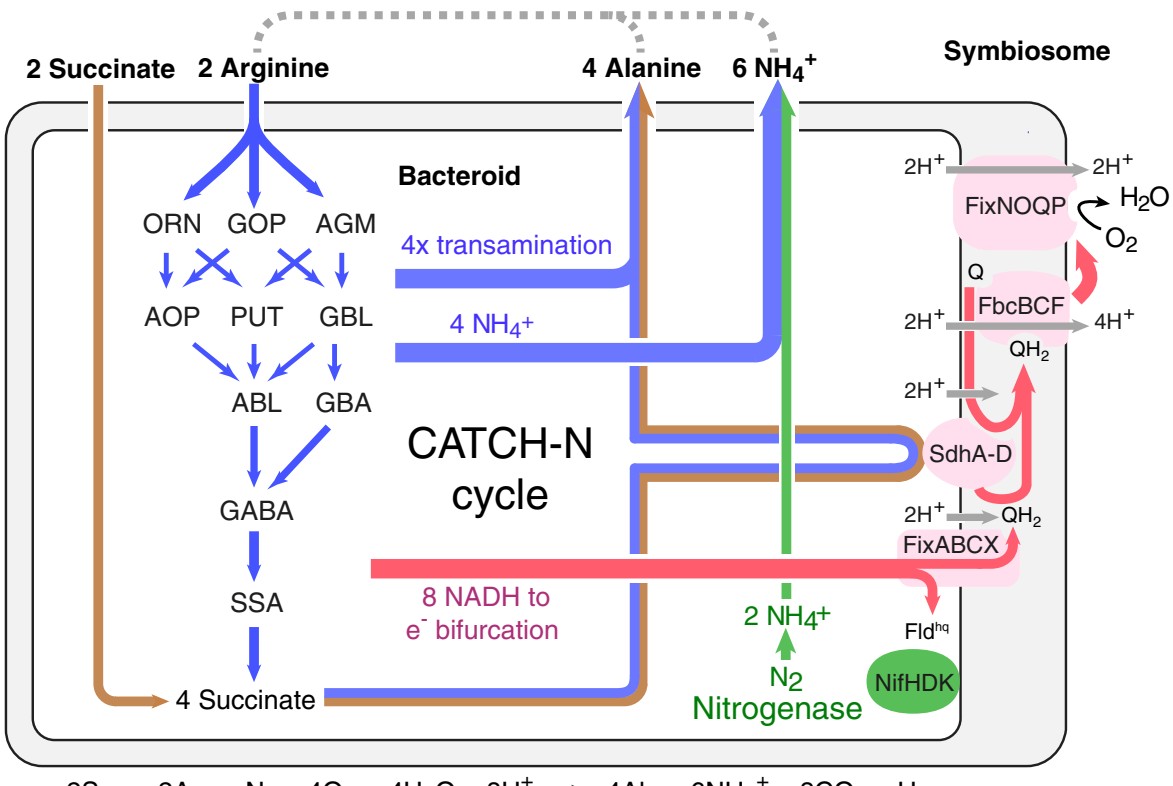

$$2Suc + 2Arg + N_2 + 4O_2 + 4H_2O + 8H^+ \longrightarrow 4Ala + 6NH_4^+ + 8CO_2 + H_2$$

**Figure 5. Model of the CATCH-N cycle operating in $N_2$-fixing bacteroids.**

Arginine (blue) and succinate (brown) are co-feed to bacteroids in equimolar ratio. Co-catabolism is interlinked through an arginine–pyruvate transamination step yielding alanine. Alternatively, an arginine–oxaloacetate transamination step yields aspartate. Enzymatic conversion of arginine releases ammonium (blue) independent from the nitrogenase reaction (green). NADH (magenta) produced through the co-catabolism is regenerated over a bifurcated electron transport chain onto terminal acceptors oxygen and nitrogen.

proton gradients are not exclusively generated by the proton expelling complexes of the electron transport chain, but moreover can also be established through proton-consuming enzymatic reactions in the cytosol, including the production of ammonia and decarboxylation reactions implemented in the CATCH-N cycle. Based on these considerations, we calculated the net proton consumption for several theoretical nitrogen fixation cycles (CATCH_N1-4) that start with arginine and succinate and feature dedicated transamination reactions that channel the TCA products pyruvate and oxaloacetate into the corresponding amino acids alanine and aspartate (Appendix). In addition to previously annotated pathways (Zhao *et al*, 2012), these theoretical nitrogen fixation cycles also integrated all enzymatic activities of the arginine transamination network identified during our biochemical enzyme studies. To evaluate the feasibility of these theoretical cycles, we estimated the net gain in ATP per $N_2$ molecule converted. Our calculations show that in terms of ATP production, the CATCH-N cycles are generally more energy-efficient than a stand-alone TCA cycle operating on succinate or malate (Table 2, Appendix) and were also more energy-efficient than the core sequence of other naturally existing arginine degradation pathways, such as the arginine–pyruvate–glutamate super-pathway (Appendix). As an example, two of our CATCH-N cycles generated on average over 18-fold more ATP per $N_2$ converted into ammonia as compared to the core sequence of the TCA cycle operating with succinate as the sole substrate. In sum, these calculations establish a new conceptional framework to understand and engineer symbiotic nitrogen-fixing organisms with future perspectives for agriculture.

## Discussion

Here, we report on the CATCH-N cycle operating on the provision of the two substrates arginine and succinate by the plant as part of a specific metabolic network that drives the process of symbiotic nitrogen fixation in rhizobia. The CATCH-N cycle shares aspects with the plant mitochondrial arginine degradation pathway (Polacco *et al*, 2013; Winter *et al*, 2015) used to liberate ammonium upon germination; however, it delivers up to 25% higher yield in nitrogen in the form of two alanines and three ammonium secreted for each co-fed arginine and succinate. Thus, from the plant's perspective, the CATCH-N cycle multiplies the nitrogen releasing capacity of arginine. On the level of bacteroids, the CATCH-N cycle provides an elegant metabolic solution for maintaining an active respiratory chain under the highly acidic and microooxic conditions present within the lysosomal compartment of the symbiosome, which is in agreement with the symbiosome-as-phagolysosome hypothesis stated by Mellor and colleagues (Mellor, 1989).

In this model, nitrogen fixation is essential for the bacteria, which otherwise have no use for the ammonium that the plant finds so valuable (Udvardi & Kahn, 1993). It addresses the question whether bacteria can opt-out of nitrogen fixation, which might allow cheaters to prosper. If neutralizing acid is what nitrogen fixation does for the bacteria, bacteria that do not fix $N_2$ will be killed as a direct consequence of their lack of cooperation, turning this mutualism into a kind of extortion. If proton consumption is the bacteroid's goal in fixing nitrogen, the plant could control nitrogen fixation by controlling input of carbon through the provision of arginine and dicarboxylate but also through its willingness to accept aspartate or alanine as products from the symbiosome and might control the efficiency of the bacterial neutralization in this way. The existence of different alternatives may be linked to the variation of symbiotic effectiveness.

Thus, the CATCH-N cycle also functions as an effective mechanism to promote the survival of bacteroids within infected plant cells. Equimolar arginine and succinate serve as substrates, and a molar ratio of nitrogen to the oxygen of 1:4 is required to operate the CATCH-N cycle. Therefore, nitrogen fixation still depends on oxygen as terminal acceptor, while harnessing elementary nitrogen as the second electron acceptor for reducing equivalents generated by the metabolism. Also, the CATCH-N cycle requires a constant flux of 8 protons into the symbiosome to maintain the pH balance of the reaction. These protons must be translocated by the action of plant ATPases as part of the symbiosome acidification process. Thus, the operation of the CATCH-N cycle depends on the presence of an active plant metabolism.

From the nitrogen balance standpoint, a feedback loop exists between the nitrogenase function of bacteroids and the availability of arginine within the host plant. Ammonium released by bacteroids is rapidly incorporated by plant cells into glutamate, glutamine, and aspartate that all serve as precursors for the biosynthesis of ornithine and subsequently for arginine occurring within chloroplasts. The output of the CATCH-N cycle results in a net gain of assimilated nitrogen that subsequently amplifies the plant's arginine biosynthesis capacity as part of a positive feedback mechanism.

As humanity faces global challenges with population growth and climate change, we need to rethink how tomorrow's agriculture will look like. Thereby, systems-biology approaches to broaden our

**Table 2.** Comparison of selected metabolic pathways for $N_2$ fixation.

| Pathway | Substrates | Products | Reaction steps | Sum NADH | Sum QH$_2$ | O$_2$ consumed per N$_2$ | Influx H$^+$ | ATP net gain |
|---------|-----------|----------|----------------|----------|-----------|--------------------------|-------------|--------------|
| CATCH-N1 | 2Suc, 2Arg | 2Ala | 9 | 8 | 4 | −4 | 56 (48 + 8) | 2.80 |
| CATCH-N2 | 2Suc, 2Arg | 2Asp | 9 | 8 | 4 | −4 | 52 (48 + 4) | 1.60 |
| CATCH-N3 | 2Mal, 2Arg | 2Ala | 9 | 8 | 1 | −3 | 44 (36 + 8) | −0.80 |
| CATCH-N4 | 2Mal, 2Arg | 2Asp | 9 | 8 | 1 | −3 | 40 (36 + 4) | −2.00 |
| TCA | 1.6Mal | CO$_2$ | 11 | 8 | 1.6 | −2.8 | 38.8 (33.6 + 5.2) | −2.76 |
| TCA | 1.6Suc | CO$_2$ | 11 | 8 | 3.2 | −3.6 | 48.8 (43.6 + 5.2) | 0.12 |

Shown are the numbers of substrate; reaction steps; and the numbers of NADH, QH2, and oxygen molecules required (negative numbers) or generated (positive numbers) during conversion of $N_2$ into two molecules of ammonia. The influx in H$^+$ for each cycle is listed with O$_2$-dependent proton translocation and O$_2$-independent proton consumption by enzymatic reactions listed in brackets. An H$^+$/ATP ratio of 3.33 was assumed for estimating the gain in ATP per $N_2$ converted.

understanding of plant–microbe interactions, as well as the design of synthetic nitrogen-fixing microbes that mimic natural symbiosis with plants, hold significant promise. Our integrated model of the CATCH-N cycle provides new insights into the principles underlying legume symbiosis and comprises an important stepping stone for the rational biotechnological engineering of artificial nitrogen-fixing microbes and improved crop plants to ensure food and climate security.

# Materials and Methods

**Reagents and Tools table**

| Reagent/Resource | Reference or source | Identifier or catalog number |
|---|---|---|
| **Experimental Models** | | |
| *B. diazoefficiens* USDA 110 spc4 | Regensburger and Hennecke (1983) | NA |
| *E. coli* BL21 *pET42b::gabT2* | This study | BC3422 |
| *E. coli* BL21 *pET42b::gabD9* | This study | BC3421 |
| *E. coli* BL21 *pET42b:: gabD6* | This study | BC4312 |
| *E. coli* BL21 *pET42b::odcB* | This study | BC4308 |
| *E. coli* BL21 *pET42b:: gabD8* | This study | BC4294 |
| *E. coli* BL21 *pET42b::argI2* | This study | BC3424 |
| *E. coli* BL21 *pET42b::speB* | This study | BC4265 |
| *E. coli* BL21 *pET42b::speB2* | This study | BC4266 |
| *E. coli* BL21 *pET42b::argD* | This study | BC4270 |
| *E. coli* BL21 *pET42b::datA* | This study | BC4292 |
| *E. coli* BL21 *pET42b::aspC* | This study | BC3417 |
| *E. coli* BL21 *pET42b::iluB1* | This study | BC3419 |
| *E. coli* BL21 *pET42b::gabD7* | This study | BC4296 |
| *E. coli* BL21 *pET42b::gabD1* | This study | BC3420 |
| *E. coli* BL21 *pET42b::odcA* | This study | BC4269 |
| *E. coli* BL21 *pET42b::argI1* | This study | BC3423 |
| *E. coli* BL21 *pET42b::aatB* | This study | BC4293 |
| *E. coli* BL21 *pET42b::gabT3* | This study | BC4271 |
| *S. meliloti* CL150 (Rm1021 *pstC*[+] *ecfR1*[+]) | Schlüter *et al* (2013) | BC2175 |
| *S. meliloti* CL150 *nifD*::Tn5-233 | Lang *et al* (2018) | CL309 |
| *S. meliloti* CL150 *dctAB::aacC1* | This study | BC4081 |
| *S. meliloti* CL150 *argI2:: aacC1* | This study | BC3455 |
| *S. meliloti* CL150 *satABC::aacC1* | This study | BC3451 |
| *S. meliloti* CL150 *ureGFE::aacC1* | This study | BC4083 |
| *S. meliloti* CL150 *aspC::aacC1* | This study | BC3457 |
| *S. meliloti* CL150 *argI1::aacC1* | This study | BC3453 |
| *S. meliloti* CL150 *argI1::smR argI2::aacC1* | This study | BC3766 |
| *S. meliloti* CL150 *artABCDE::aacC1* | This study | BC3459 |
| *S. meliloti* CL150 *amtB::aacC1* | This study | BC4085 |
| *M. truncatula* Jemalong *wt* | Pecrix *et al* (2018) | NA |
| *M. truncatula* Jemalong *lss* | Schnabel *et al* (2010) | NA |
| *G. max* cultivar Williams | | NA |
| **Oligonucleotides and sequence-based reagents** | | |
| PCR primers | This study | Dataset EV5 |
| **Chemicals, enzymes and other reagents** | | |
| 4-Aminobutanoic acid | Sigma-Aldrich | A2129-10G |

## Reagents and Tools table   (continued)

| Reagent/Resource | Reference or source | Identifier or catalog number |
|---|---|---|
| 4-Guanidinobutyric acid | Sigma-Aldrich | G6503-1G |
| Agmatine | Sigma-Aldrich | 101443-1G |
| Aminobutanal | This work | NA |
| Guanidinobutanal | This work | NA |
| L-Arginine | Fluka | 11010 |
| L-Arginine (13C) | Sigma-Aldrich | 643440-100MG |
| L-Arginine (15N) | Sigma-Aldrich | 643440-100MG |
| L-Citrulline | Sigma-Aldrich | C7629-1G |
| L-Ornithine | Sigma-Aldrich | O8305-25G |
| NAD$^+$ | Sigma-Aldrich | N1636 |
| Putrescine | Sigma-Aldrich | P5780 |
| Pyruvate | Sigma-Aldrich | P2256-100G |
| Succinate | Sigma-Aldrich | 150-90-3 |
| Succinic semialdehyde | This work | NA |
| Sulfuric acid | Sigma-Aldrich | 258105 |
| Sodium hypochlorite solution | VWR | BDH7038 |
| Restriction enzyme SpeI | New England Biolabs | R0133 |
| Restriction enzyme MfeI | New England Biolabs | R0589 |
| **Other** | | |
| GC6850 gas chromatograph instrument | Agilent Technologies | |
| 6550 accurate-mass quadrupole time-of-flight | Agilent Technologies | |
| Agilent HILIC Plus RRHD column | Agilent Technologies | |
| 5500 QTRAP triple-quadrupole mass spectrometer | AB Sciex | |
| HisTrap FF crude column | GE Healthcare | GE11-0004-58 |

## Materials and Methods

### Bacterial strains, cultivation, and growth conditions

*Sinorhizobium meliloti* strain CL150 (Rm1021, *pstC*$^+$ *ecfR1*$^+$) (Schlüter *et al*, 2013) was grown at 30°C in LB broth medium with 5 g NaCl per liter. *Escherichia coli* was grown in LB broth at 37°C. Where necessary, growth media were supplemented with antibiotics at the following concentrations: gentamicin, 10 μg/ml for *E. coli* and 30 μg/ml for *S. meliloti* when cultured in LB; streptomycin, 200 μg/ml; and ampicillin, 50 μg/ml.

### Plant cultivation and inoculation assays

*Medicago truncatula* WT Jemalong seeds were surface-sterilized with 70% ethanol for five minutes and thoroughly rinsed with water. Seeds were imbibed with gentle agitation for at least four hours with two water changes and further imbibed overnight at room temperature without light. After imbibition, seeds were washed with water and germinated at 30°C for 24 h. Seedlings were planted into a sterile perlite substrate (Isoself, Knauf) within 300 cm$^3$ black plastic pots (Greenhouse, Elho). Plants were grown in plant growth shelves at room temperature with a controlled 16-h day and 8-h night cycle. During the light cycle, every pot received 2500 lumens using 36 W Fluora 77 OSRAM light bulbs. Plants were automatically watered with a droplet watering system (Micro-drip-system, Gardena Art.

8311-20 Art. 1407-20) at 3-day intervals with 80 ml of 10% BNM solution (Ehrhardt *et al*, 1992). Pots were covered with a transparent PET cylinder during growth. Three days after germination, plants were inoculated with 20 ml of a *S. meliloti* cell culture with a cell density of an OD$_{600nm}$ of 0.05 resuspended in 1 mM MgSO$_4$. The inoculation time point was considered as day zero post-inoculation (dpi).

*Glycine max* cultivar Williams seeds were surface-sterilized with a wash in 100% ethanol for 5 min followed by a wash in 35% H$_2$O$_2$ for 15 min. Afterward, seeds were thoroughly rinsed with water. Seeds were germinated in 0.8% agar plates for 24–48 h at 25°C in the dark. Seedlings were planted on 200 cm$^3$ brown glass jars filled with autoclaved vermiculite (Vermisol) and 100 ml of Jensen media (Vincent, 1970). After transplantation, plants were inoculated with *B. diazoefficiens* (1 ml OD$_{600\ nm}$ of 0.01). The inoculation time point was considered as day zero post-inoculation (dpi). Plants were grown in plant growth chambers at 28°C with a controlled 16-h day and 8-h night cycle as previously described (Göttfert *et al*, 1990). After 5 dpi, plants were watered every 3 days with sterile water.

### Phenotypic characterization of plants to assess symbiotic nitrogen fixation

Symbiosis phenotypes such as nodulation frequency, plant dry weight, and nitrogenase activity were determined in *M. truncatula* WT plants 8 weeks post-inoculation with *S. meliloti* gene deletion

mutants and wild-type control strains. Nitrogenase activity of *M. truncatula* was determined by measuring acetylene reduction to ethylene as previously reported. Plants were placed into 50 cm$^3$ sealed vials to which acetylene was injected to a final concentration of 2%. Ethylene production was measured after 3 h and 5 h of incubation on an analytic gas chromatograph instrument (GC6850, Agilent Technologies). The ethylene background was monitored and systematically removed from every measurement. To determine the average ethylene production per nodule, ethylene production was normalized by nodule number and duration of acetylene incubation. Nodulation frequency was determined by manually counting all the visible nodules per plant. For dry weight determination, plant shoots were dried at 85°C for at least 20 h prior individual plant weight measurement (ABS 80-40 N, KERN). Measurements from replicas were averaged and normalized to the *S. meliloti* CL150 WT reference strain. Nitrogenase activity, plant dry weight, and nodulation frequency were determined for at least 22 independent plants for each assayed *S. meliloti* strain.

## Sinorhizobium meliloti *bacteroid ultrastructure determination through scanning electron microscopy*

*Medicago truncatula* nodules were harvested 10 weeks post-inoculation (wpi), cut open longitudinally, and collected in a vessel with fixative buffer (2.5% glutaraldehyde, 2% formaldehyde in 0.15 M Na-cacodylate with 2 mM CaCl$_2$). The nodule containing vessel was subjected to vacuum for at least 30 min to allow the nodules to sink followed by microwave fixation in fresh fixative (PELCO BioWave Pro$^+$) and washed with fixative buffer. Nodules were stained through sequential incubation with (i) 2% OsO$_4$, 1.5% K$_4$Fe(CN)$_6$ in 0.15 M Na-Cacodylate with 2 mM CaCl$_2$; (ii) 1% thiocarbohydrazide; (iii) 2% OsO$_4$; (iv) 1% uranylacetate; and (v) Waltons lead aspartate; samples were washed with fixative buffer between every step. Samples were dehydrated through a graded ethanol series (25%, 50%, 75%, 100%, dry ethanol) and washed with dry acetone. Samples were placed in a graded series of Epon/Araldite in dry acetone (25%, 50%, 75%, 100%, 100%) and allowed to polymerize at 60°C for 3 days. Ultrathin sections (100 nm) were transferred to Si-wafer-chips and imaged on a FE-SEM FEI Magellan 400i operating at 1.8 kV and 0.8 nA with a 20 nm pixel size by backscatter electron detection.

## Substrate specific stimulation of nitrogenase activity in isolated bacteroids

*Sinorhizobium meliloti* and *B. diazoefficiens* bacteroids were isolated from *M. truncatula* and *G. max* root nodules, respectively, according to following procedure. Plant roots were harvested 3 weeks (*G. max*) or 10 weeks (*M. truncatula*) post-inoculation with wild-type *B. diazoefficiens* 110spc4 strain (Regensburger & Hennecke, 1983) and *S. meliloti* CL150 strain, respectively, and rinsed thoroughly with water. Under anaerobic conditions (92% N$_2$, 8% H$_2$), nodules (250 mg to 1 g wet weight) were crushed in PBS (10 mM Na$_2$HPO$_4$, 1.8 mM KH$_2$PO$_4$, 137 mM NaCl, 2.7 mM KCl, pH 7.4) and filtered through three layers of gauze to remove debris. Bacteroid suspensions (2 ml corresponding to 4 × 10$^6$ bacteroid cells) were added to 15-ml sealed flasks to obtain reference measurements of nitrogenase activity in isolated bacteroids in the presence of plant crude extract. To assay stimulation of nitrogenase activities upon addition of substrates, bacteroid suspensions

were washed two times with PBS under anaerobic conditions and pelleted by centrifugation at 1,000 *g*. Recovered bacteroids (4 × 10$^6$ cells) were resuspended in 2 ml of induction media (2 μM biotin, 1 mM MgSO$_4$, 42.2 mM Na$_2$HPO$_4$}, 22 mM KH$_2$PO$_4$, 8.5 mM NaCl, 21 nM CoCl$_2$, 1 μM NaMoO$_4$ pH 7.0) and added to 15-ml sealed flasks. Induction media were supplemented with either 7.4 mM succinate or 5 mM arginine or both substrates. Acetylene and oxygen were added to a final concentration of 5% and 0.01%, respectively, in the head space of each flask. Nitrogenase activity was determined for each sample through measuring the reduction of acetylene into ethylene after 1 and 3 h time points. Ethylene production was detected with gas chromatograph (GC6850, Agilent Technologies). Activities were normalized by nodule wet weight. Furthermore, relative activities were calculated by subtracting the baseline activity of washed bacteroid samples without substrate supplementation. Reported *B. diazoefficiens* and *S. meliloti* activity is the result of 17 and 4 independent preparations, respectively.

## Substrate specific stimulation of ATP production in isolated bacteroids

*Bradyrhizobium diazoefficiens* bacteroids were isolated from 3 weeks post-inoculated *G. max* root nodules. Nodules (1 g wet weight) were crushed in PBS (10 mM Na$_2$HPO$_4$, 1.8 mM KH$_2$PO$_4$, 137 mM NaCl, 2.7 mM KCl, pH 7.4) and filtered through three layers of gauze to remove debris. Bacteroid suspension was pelleted by centrifugation at 2,500 *g,* and the supernatant (nodule extract) was saved for later usage. Bacteroid suspension (5 × 10$^8$} cells) was washed twice with PBS and resuspended in 1 ml of induction media (2 μM biotin, 1 mM MgSO4, 42.2 mM Na$_2$HPO$_4$, 22 mM KH$_2$PO$_4$, 8.5 mM NaCl, 21 nM CoCl$_2$, 1 μM NaMoO$_4$ pH 7.0). To avoid ATP generation from aerobic respiration, the bacteroid suspension was placed under anaerobic conditions (92% N$_2$, 8% H$_2$) to perform the rest of the procedure. Bacteroids (200 μL) were incubated in 2 mL of nodule extract (obtained at the beginning of the protocol) or induction media without supplements or supplemented with either 7.4 mM succinate or 5 mM arginine or both substrates. ATP content was determined for each sample (using a 1:10 dilution) through ATP-dependent luciferase reaction (BacTiter-Glo Microbial Cell Viability Assay, Promega). Luminescence from luciferase activity was quantified with Victor3 multilabel plate counter (PerkinElmer).

## Transposon mutagenesis

To generate hyper-saturated transposon mutant libraries in *S. meliloti*, a previously described Tn5 mutagenesis procedure for *Caulobacter crescentus* was adapted (Christen *et al*, 2011). In brief, the Tn5 delivery ColE1 plasmid pTn5_gent_14N (Christen *et al*, 2016) was conjugated from an *E. coli* SM10 donor strain into a *S. meliloti* CL150 recipient strain. Separate transposon mutant libraries were generated and growth selected on rich medium (LB). For each condition, a total of sixteen independent conjugations were performed and replicate libraries were tagged using eight barcoded Tn5 derivatives. Transposon insertion mutants were selected on LB supplemented with gentamicin and streptomycin. Plates were incubated at 30°C for 2 days, and transposon mutant libraries from each plate were separately pooled, supplemented with 10% v/v DMSO (Sigma-Aldrich), and stored in 96-well deep-well plates at −80°C for further processing.

## In planta *selection of transposon mutant libraries*

Transposon mutant pools were selected for infection and nodule formation in legume plants. For nodulation experiments, a *Medicago truncatula* lss super-nodulator mutant was used (Schnabel *et al*, 2010). Seeds were treated with concentrated $H_2SO_4$ (Sigma-Aldrich) for 5 min, thoroughly rinsed with sterile water, then sterilized with 7% NaClO (VWR chemicals) for 3 min, and again rinsed with sterile water. The seeds were then imbibed with gentle agitation for four hours with regular water changes and then incubated overnight in the dark at room temperature. After imbibition, seeds were rinsed with sterile water, placed in deep petri plates, and inverted for 24 h at 30°C to allow for the downward growth of the seedling roots. After removing seed coats, groups of 25 seedlings were planted on large square plates (Genetix) containing 1.2% buffered nodulation medium (Ehrhardt *et al*, 1992) supplemented with 0.1 nM aminoethoxy vinyl glycine. Altogether, 4,500 *M. truncatula* lss plant seedlings were grown at 22°C with a controlled 16-h day and 8-h night cycle (2500 Lumens using Osram Fluora L36 W/77 bulbs). Five days post-germination, *M. truncatula* lss seedlings were flood-inoculated with *S. meliloti* Tn5 mutant reference libraries initially selected on rich media conditions (LB). During inoculation experiments, input libraries were kept independent from each other. *S. meliloti* transposon input mutant pools were inoculated from 96-well storage plates, grown overnight, washed, and resuspended to an $OD_{600 \text{ nm}} = 0.05$ in 10 mM $MgSO_4$. Plant roots were then aseptically inoculated with 5 ml of a dilute bacterial suspension, followed by removal of the excess bacterial suspension.

### Recovery of transposon mutant libraries from nodules

After 6 weeks post-inoculation, nodules were harvested. To recover transposon mutants capable of infecting root nodules, a two-step surface sterilization protocol was employed. Nodule material was washed with 1% SDS and then treated with 70% ethanol for 5 min, followed by rinsing with sterile water. Next, nodules were treated for 3 min with 0.2% NaClO (VWR chemicals) followed by three rinses with sterile water. The surface-sterilized nodules were crushed in cold PBS (10 mM $Na_2HPO_4$, 1.8 mM $KH_2PO_4$, 137 mM NaCl, 2.7 mM KCl, pH 7.4) and filtered through three layers of gauze to remove debris. The filtered suspension containing bacteroids was plated on LB supplemented with gentamicin and strepto-mycin and grown for 2 days at 30°C. The recovered *S. meliloti* colonies were pooled and arrayed in 96-well plates and stored at −80°C for subsequent use.

### Data analysis and mapping of transposon insertion sites

Raw sequencing data processing and read alignment were performed using a custom sequence analysis pipeline based on Python, Biopython (Cock *et al*, 2009), bwa (Li & Durbin, 2010), and MATLAB routines as previously described (Christen *et al*, 2016). Adapter sequences were detected using Python string comparison with a 15 bp search window. Demultiplexing into the different TnSeq selection experiments was performed according to a defined barcode sequence tag internal to the arbitrary primer sequence. Reads were aligned onto the *S. meliloti* 1021 NCBI reference genome (NC_003047, NC_003037, NC_003078; Barnett *et al*, 2001; Finan *et al*, 2001; Galibert *et al*, 2001) using bwa-07.12 (Li & Durbin, 2010). Insertion datasets were correlated with the genome annotation to analyze global insertion statistics and calculate transposon insertion occurrence and distributions within each annotated GenBank feature of the *S. meliloti* 1,021 genome (Galibert *et al*, 2001)} as previously described (Christen *et al*, 2016).

### Gene essentiality analysis across selection conditions

The metrics and statistical analysis for classification of protein-coding sequences (CDS) into essential, promoting fitness, and non-essential categories have been previously described (Christen *et al*, 2011). CDS are classified as "essential" or "fitness" genes according to the following criteria. CDS are classified as essential if the transposon insertion density is < 6 times the average insertion density measured across the genome and has either a single transposon insertion gap covering more than 60% of the CDS or has two internal transposon gaps covering more than 80% of the CDS. If the average transposon density is less than four times the average insertion density across the genome but no essentiality criterion was satisfied, CDS were classified as fitness genes. For the nodule selected TnSeq dataset, a symbiosis impairment score (*P*-values) was calculated for each gene based on the number and distribution of transposon insertions recovered from the *in planta* selection experiments. The symbiosis impairment score was calculated for each gene as the product of the probability p1, to recover *i* transposon insertions when λ insertions are expected according to a Poisson distribution, and the probability p2, to observe a consecutive loss in transposon insertions equals or exceeding k insertions as compared to the input mutant pool. The expected number of recovered insertions was estimated based on the observed loss rate of neutral insertions ($l = 6.840$) due to sampling. The probability p1 was calculated according to the following formula:

$$p1 = \sum_{s=0}^{i} \frac{(e^{-\lambda}) \cdot (\lambda^s)}{s!}$$

The probability p2 was calculated for each gene based on the total number of insertions (n) present in the input mutant pool, the number of recovered insertions from the output mutant pool (i), and the maximal number of consecutive insertions that where lost upon *in planta* selection (k) by calculating the number (h) of compositions of length i + 1 for n where each part does not exceed k, divided by the total number of compositions ($h_{tot}$) for n. The probability p2 was calculated according to the following formula:

$$h = \frac{(i+1) \cdot (n-k)!}{(i)! \cdot (n-i-k)!} \quad h_{tot} = \frac{n!}{(i)! \cdot (n-i)!} \quad p_2 = \frac{h}{h_{tot}} \text{ for } k > \frac{n-i}{2}$$

For k smaller than (n−i)/2 and n not to exceed 100, p2 was numerically calculated by calculating the number (h) of compositions of length i + 1} for n where each part does not exceed k by brute force. For k smaller than (n−i)/2 and n equals or exceeding 100, we approximated p2 by sampling the composition space by random simulation and counting the occurrences of compositions of length i + 1 for n where each part does not exceed k.

### Construction of targeted gene deletions in Sinorhizobium meliloti

*Sinorhizobium meliloti* deletion mutants were generated by replacing the native gene with the *aacC1* gene conferring gentamycin resistance using a one-step double homologous recombination procedure as detailed in Ledermann *et al* (2016). Flanking DNA

regions covering 750 bp upstream and downstream of a target gene were PCR amplified (Dataset EV5) and subsequently fused to a central gentamicin resistance gene using splicing by overlapping extension PCR or Gibson assembly (Gibson *et al*, 2009) to produce gene replacement cassettes. The gene replacement cassettes were cloned via SpeI and MfeI into the pNPTS138 plasmid, which is non-replicative in *S. meliloti* and confers kanamycin resistance. Cloned plasmids were sequence confirmed and conjugated into *S. meliloti* strain CL150. Recombinants were selected on LB media supplemented with gentamicin and subsequently screened for kanamycin sensitivity. Single gene deletion strains were confirmed by PCR and sequencing. A similar deletion strategy was employed for construction of the double arginase mutant (Δ*argI1* Δ*argI2*) with a gene replacement cassette for *argI1* carrying a spectinomycin marker gene. The resulting plasmid was conjugated into *S. meliloti* CL150 Δ*argI2*, and the double arginase double mutant strain was screened for kanamycin sensitivity before verification of constructed deletion by PCR and sequencing.

### $^{13}C$ arginine isotope tracing in Bradyrhizobium diazoefficiens bacteroids

*Bradyrhizobium diazoefficiens* bacteroids were isolated under anaerobic conditions from three weeks post-inoculated *G. max* root nodules according to Sarma and Emerich (2005) and Delmotte *et al* (2010). Nodules (10 g wet weight) were crushed in PBS (10 mM $Na_2HPO_4$, 1.8 mM $KH_2PO_4$, 137 mM NaCl, 2.7 mM KCl, pH 7.4). The homogenate was passed through four layers of cheesecloth (pre-moistened with PBS) into a sterile centrifuge tube, re-extracted several times with buffer, and centrifuged at 400 *g* for 10 min. The resulting pellet was resuspended twice in PBS and centrifuged at 8,000 *g* for 20 min. The pellet was dispersed into the extraction buffer (2 ml/g original weight of the nodule) and was layered onto a pre-equilibrated Ludox gradient consisting of 25% (10 ml) Ludox and 75% PBS (30 ml). The gradient tubes were centrifuged in an SW-28 rotor at 10,000 *g* for 35 min at 4°C in a Beckman L8–55 ultracentrifuge. The bacteroid layer with a density of 1.09 g/ml was collected. The bacteroid pellet was suspended in distilled water and collected by centrifugation at 10,000 *g*. For labeling assays, bacteroids were resuspended in 10 ml volume in PBS to a final OD of 4.0. After the addition of succinate (5 mM) and $^{13}C$ arginine (5 mM), bacteroids were incubated at RT under microaerobic condition (0.1% v/v) and samples of 250 μl were taken at regular time intervals, filtered on a PVDF 0.45 μl membrane and immediately washed with 1.0 ml $H_2O$ Chromasolv. The filter with the cell pellet was extracted in 3 ml (40% MeOH, 40% acetonitrile and 20% $H_2O$) at −20°C for 1 h, stored at −80°C, and finally dried in a SpeedVac. The metabolite extracts were resuspended in 100 μl MilliQ water, and metabolites were analyzed using a HILIC method. 5 μl of metabolite extract was injected on an Agilent HILIC Plus RRHD column (100 mm × 2.1 mm × 1.8 μm; Agilent). The gradient of mobile phase A (10 mM ammonium formate and 0.1% formic acid) and mobile phase B (acetonitrile with 0.1% formic acid) was as follows: 0 min, 90% B; 2 min, 40% B; 3 min, 40% B; 5 min, 90% B; and 6 min, 90% B. The flow rate was held constant at 400 μl $min^{-1}$. Metabolites were detected on a 5500 QTRAP triple-quadrupole mass spectrometer in positive mode with MRM scan type (AB Sciex, Foster City, CA). The raw data were processed and analyzed by custom software using MATLAB (MathWorks).

### $^{15}N$ arginine isotope tracing in Bradyrhizobium diazoefficiens bacteroids

*Bradyrhizobium diazoefficiens* bacteroids were isolated from 3 weeks post-inoculated *G. max* root nodules under anaerobic conditions as described above. Bacteroids were resuspended (1 ml/g nodule wet weight) in 2 ml of tracing media (2 μM biotin, 1 mM $MgSO_4$, 42.2 mM $Na_2HPO_4$, 22 mM $KH_2PO_4$, 8.5 mM NaCl, 21 nM $CoCl_2$, 1 μM $NaMoO_4$ pH 7.0, 10 mM $NH_4Cl$, 7.4 mM succinate, and 5 mM $^{15}N$ arginine). Bacteroid suspensions were incubated at room temperature under microaerobic conditions (0.1% v/v). Aliquots (20 μl) of the enzymatic reaction were sampled over the time series, and reaction was blocked by adding 180 μl of ice-cold methanol. Relative metabolite abundances were determined by non-targeted flow injection analysis as described previously (Fuhrer *et al*, 2011). Mass spectra were recorded in negative-ionization profile mode from m/z 50 to m/z 1,000 on an Agilent 6550 accurate-mass quadrupole time-of-flight instrument with a frequency of 1.4 spectra/s using the highest resolving power (4 GHz HiRes). The source gas temperature was set to 225°C, with 11 l $min^{-1}$ drying gas and a nebulizer pressure of 20 psig. The sheath gas temperature was set to 350°C, and the flow rate was 10 l $min^{-1}$. Electrospray nozzle and capillary voltages were set at 2,000 and 3,500 V, respectively.

### Purification of recombinant proteins

Coding sequences of interest were amplified by PCR (Dataset EV5) and cloned into pET42 expression plasmids inframe to a C-terminal (His)6-tag (6xHis). The resulting vectors were sequence-verified and electroporated into BL21 rosetta pLys strains. *E. coli* BL21 harboring the expression vectors were grown at 30°C in LB medium containing chloramphenicol (20 mg/l) and kanamycin (30 mg/l). When the cultures reached an $OD_{600nm}$ of 0.4, isopropyl-β-D-thiogalactopyranoside (IPTG) was added to a final concentration of 0.5 M. After the addition of IPTG, the culture was grown for 2–4 h more at 30°C to induce the expression of the recombinant proteins. Grown cells were harvested by centrifugation at 5,095 *g* for 10 min at 4°C. The resulting pellet was either shock frozen with liquid $N_2$ and stored at −80°C or immediately used for protein extraction and purification. To purify the recombinant proteins, cells were resuspended in NPI-buffer (20 mM sodium phosphate, 500 mM NaCl, pH7.4) supplemented with 10 mM imidazole and disrupted using a French press (SLM Instruments Inc.). Lysates were cleared by centrifugation at 17,000 *g* for 15 min at 4°C to remove cell debris. The supernatant was loaded on HisTrap FF crude column (GE Healthcare) previously equilibrated with NPI-buffer supplemented with 10 mM imidazole. The column was washed twice with NPI-buffer supplemented with 10 mM and 20 mM imidazole, respectively. The purified enzymes were eluted with NPI-buffer supplemented with 250 mM imidazole, concentrated by ultrafiltration with Amicon Ultra-4 centrifugal filters (Merck Millipore), and dialyzed against NPI-buffer containing 10% (v/v) glycerol. Protein concentration was determined using the Pierce BCA Protein Assay kit (Thermo Scientific). Protein samples were shocked frozen with liquid $N_2$ and stored at −80°C until use.

### Chemical synthesis of succinate semialdehyde, 4-aminobutanal, and 4-guanidinobutanal

The arginine transamination metabolic network contains a set of 13 intermediates. Thereof, three intermediates succinate semialdehyde, 4-aminobutanal, and 4-guanidinobutanal are not commercially

available and were chemically synthesized for use in subsequent enzyme characterization studies as aldehyde dehydrogenase substrates.

Succinate semialdehyde was synthesized according to the procedure as previously described (Bruce *et al*, 1971). In a 15 ml Falcon tube, a solution of monosodium glutamate (169 mg, 1 mmol) in 5.0 ml distilled water was exposed to a gentle flow of $N_2$ for 5 min. An equimolar amount of chloramine T (227 mg, 1 mmol) was added to the solution and dissolved by heating the solution to 60°C. After further incubation at 60°C for an additional 15 min, the mixture was cooled to 25°C and adjusted to pH 2.0 with concentrated HCl and degassed. Reaction by-products were crystallized by placing the mixture on ice and removed by filtration. The aqueous phase was extracted with diethyl ether 3 times, organic phases were combined, and the water fraction was discarded. Diethyl ether was evaporated yielding an aqueous solution of 1% of the starting material and a concentration of approximately 2 M succinic semialdehyde.

4-guanidinobutyraldehyde was prepared from L-arginine according to Tanaka *et al* (2001). The procedure follows a similar reaction scheme as detailed for the synthesis of succinic semialdehyde with the following modifications. As starting material, arginine hydrochloride (210 mg, 1 mmol) was dissolved. Upon addition of chloramine T (227 mg, 1 mmol), the mixture was adjusted to a pH 6.5 with 1 N HCl (50 μl) and heated to 60°C. Prior extraction with diethyl ether, the solution was adjusted to pH 13.5 with 10 N NaOH.

4-aminobutyraldehyde was prepared by the hydrolysis of 1.0 ml of 0.5 M 4-aminobutyraldehyde diethyl acetal (Sigma-Aldrich). 172 μl 4-aminobutyraldehyde diethyl acetal was dissolved in 2 ml $H_2O$, and 1 N HCl was added to acidify pH 3.0. After incubation for 30 min at 30°C, the reaction mixture was titrated to pH 10.0 with 1 N NaOH. The crude reaction product 4-aminobutanal was extracted from the water phase with five consecutive extraction steps using 1 ml diethyl ether. Ether fractions were combined, and the solvent was removed by evaporation. The reaction product 4-aminobutyraldehyde was obtained as a colorless liquid of 40.0 mg mass.

### Enzyme characterization and activity assays

Transaminases were assayed for enzymatic activity according to the following procedure. A total of 30 μg purified enzyme were added to 200 μl reaction mixture containing 1 mM arginine, 1 mM ornithine, 1 mM citrulline, 1 mM agmatine, 1 mM putrescine, 1 mM 4-guanidinobutanoate, 1 mM 4-aminobutanoate, 10 mM of pyruvate, 1 mM $MgCl_2$, 1 mM $MnCl_2$, and 100 μM pyridoxal phosphate in 50 mM PBS pH 7.4 followed by incubation at 25°C. Aliquots (20 μl) of the enzymatic reaction were sampled over the time series, and reactions were blocked by adding 180 μl of ice-cold methanol. Ureohydrolases were assayed for enzymatic activity according to the following procedure. A total of 30 μg purified enzyme were added to 200 μl reaction mixture containing 1 mM arginine, 1 mM agmatine, 1 mM 4-guanidinobutanoate, 10 mM of pyruvate, 1 mM $MgCl_2$, 1 mM $MnCl_2$, and 100 μM pyridoxal phosphate in 50 mM PBS pH 8.0, previously incubated at 25°C for 30 min with 0.15 ng/μl purified AspC followed by heat inactivation at 65°C for 5 min to generate 5-guanidino-2-oxopentanoate (GOP) and guanidinobutanal, from arginine and agmatine transamination, respectively. Ureohydrolase reactions were done at 25°C. Aliquots

(20 μl) of the enzymatic reaction were sampled over the time series, and reaction was blocked by adding 180 μl of ice-cold methanol. Decarboxylases were assayed for enzymatic activity according to the following procedure. A total of 30 μg purified enzyme were added to 200 μl reaction mixture containing 1 mM arginine, 1 mM ornithine, 1 mM citrulline, 10 mM of pyruvate, 1 mM $MgCl_2$, 1 mM $MnCl_2$, 100 μM pyridoxal phosphate, and 500 μM thiamine pyrophosphate in 50 mM PBS pH 8.0, previously incubated at 25°C for 30 min with 0.15 ng/μl purified AspC followed by heat inactivation at 65°C for 5 min to generate 5-guanidino-2-oxopentanoate (GOP) from arginine transamination. Decarboxylase reactions were done at 25°C. Aliquots (20 μl) of the enzymatic reaction were sampled over the time series, and reactions were blocked by adding 180 μl of ice-cold methanol. Substrate consumption and product formation of transaminases, ureohydrolases, and decarboxylases reactions were determined by non-targeted flow injection MS analysis as described previously (Fuhrer *et al*, 2011). Dehydrogenase activities were assayed from cell lysates of BL21 rosetta pLys strain expressing *S. meliloti* dehydrogenases according to the following procedure. Cell lysates, corresponding to 20 μg of dehydrogenase enzymes, were added to a substrate mixture containing 1 mM succinate semialdehyde or 1 mM 4-guanidinobutanal or 1 mM 4-aminobutanal in 10 mM PBS pH 10.0 supplemented with 1 mM $NAD^+$. Dehydrogenase reaction was determined by the conversion of $NAD^+$ into $NADH + H^+$, which was measured by the increase in absorbance at 340 nm.

### Biochemical reconstitution of the catabolic arginine transamination network

To reconstruct a functional catabolic arginine transamination network *in vitro*, a multienzyme assay comprising 14 purified enzymes was established. A reaction buffer containing of 500 μM thiamine pyrophosphate, 100 μM pyridoxal phosphate, 2 mM NAD, 1 mM $MgCl_2$, and 1 mM $MnCl_2$ in 50 mM PBS pH 8.0 was prepared. Purified enzymes (25 μg each) from the catabolic arginine transamination network (AspC, AatB, ArgD, GabT2, DatA, ArgI1, SpeB, SpeB2, IlvB1, OdcA, OdcB, GabD1, GabD6, and GabD7) were added one by one and gently mixed into the reaction mixture. After the addition of all enzymes, 20 mM pyruvate and 2 mM arginine were added and gently mixed. As a control, the enzyme mix was incubated in a reaction buffer lacking arginine and pyruvate. Aliquots (15 μl) of the enzymatic reaction were sampled over the time series, and reaction was blocked by adding 135 μl of ice-cold methanol. Substrate consumption and product formation of enzymatic reactions were determined by non-targeted flow injection MS analysis as described previously (Fuhrer *et al*, 2011).

## Data availability

The datasets produced and presented in this study are available as Datasets EV1–EV5.

**Expanded View** for this article is available online.

## Acknowledgments

We thank R. Schlapbach and L. Poveda from ZFGC and Christa Pennacchio from the Joint Genome Institute for sequencing support, Anne Greet

Bittermann and members from ScopeM, ETH Zürich for electron microscopy support, and H. Christen for the conception of computational algorithms, mathematical conception, and the statistical framework for TnSeq data analysis. We thank H-M. Fischer and R. Ledermann for help with plant phenotypic analyses and nitrogenase assays and H. Hennecke for helpful discussions and comments on the manuscript. This work received institutional support from the Swiss Federal Institute of Technology (ETH) Zürich, ETH research grant (ETH-08 16-1) to B.C, the Swiss National Science Foundation (31003A_166476, 310030_184664 and CRSII5_177164) to B.C, and a Community Science Program (CSP) DNA sequencing award (CSP-1107) from the U.S. Department of Energy Joint Genome Institute in Walnut Creek. CA, USA. The work conducted by the U.S. Department of Energy Joint Genome Institute, a DOE Office of Science User Facility, is supported by the Office of Science of the U.S. Department of Energy under Contract No. DE-AC02-05CH11231.

## Author contributions

CEF-T, FT, MC, and BC performed transposon mutagenesis experiments; CEF-T, FT, CM, MC, and BC performed research; TF performed MS measurement; BC, MC, and CEF-T performed data analysis; US and TF provided comments on the manuscript; MC and BC conceived the theoretical concept of the CATCH-N cycle; and CEF-T, MC, and BC wrote the manuscript.

## Conflict of interest

The authors declare the following competing interests: B.C. and M.C. hold shares in Gigabases Switzerland AG, which develops microbes with improved nitrogen fixation to enhance the sustainability of agriculture.

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
