## [Review Process File · Molecular Systems Biology]

Co-catabolism of arginine and succinate drives symbiotic nitrogen fixation

Carlos Flores-Tinoco, Flavia Tschan, Tobias Fuhrer, Celine Margot, Uwe Sauer, Matthias Christen, and Beat Christen

DOI: <https://doi.org/10.15252/msb.20199419>

Corresponding author(s): Beat Christen Contributing Author Matthias Christen with email address matthias.christen@imsb.biol.ethz.ch has been identified as a secondary point of correspondence if this manuscript is accepted for publication.

Review Timeline:

Submission Date:	17th Dec 19
Editorial Decision:	12th Feb 20
Revision Received:	2nd Apr 20
Editorial Decision:	6th Apr 20
Revision Received:	13th Apr 20
Accepted:	21st Apr 20

Editor: Maria Polychronidou

Transaction Report:

(Note: With the exception of the correction of typographical or spelling errors that could be a source of ambiguity, letters and reports are not edited. The original formatting of letters and referee reports may not be reflected in this compilation. Depending on transfer agreements, referee reports obtained elsewhere may or may not be included in this compilation.)

12th Feb 2020

Manuscript Number: MSB-19-9419

Title: Co-catabolism of arginine and succinate drives symbiotic nitrogen fixation

Dear Prof. Christen,

Thank you again for submitting your work to Molecular Systems Biology. We have now heard back from the three referees who agreed to evaluate your study. As you will see below, the reviewers are quite positive and acknowledge that the presented findings are novel and relevant to a broad audience. They raise however a series of concerns, which we would ask you to address in a revision.

I think that the reviewers' recommendations are clear and I therefore see no need to repeat the points listed below. Please feel free to contact me in case you would like to discuss in further detail any of the issues raised by the reviewers.

Related to a comment by reviewer #1: our Article format does not allow combining Results and Discussion, and we would therefore ask you to keep the two sections separate. Please make sure that discussions on the main findings are mainly provided in the Discussion section and not in the Results.

On a more editorial level, we would like to ask you to address the following issues:

- Please provide a .doc version of the manuscript text (including legends for main figures, EV figures and tables) and individual files for the main figures and EV figures.
- We have replaced Supplementary Information by the Expanded View (EV format). In this case, all additional figures can be included in a PDF called Appendix. Appendix figures (and Tables, if any) should be labeled and called out as: "Appendix Figure S1, Appendix Figure S2... Appendix Table S1..." etc. Each legend should be below the corresponding Figure/Table in the Appendix. Please include a Table of Contents in the beginning of the Appendix. For detailed instructions regarding expanded view please refer to our Author Guidelines: .
- The four Datasets should be provided as EV Datasets (Dataset EV1, Dataset EV2 etc.). Please include in each .xls file a separate tab with a brief description of the Dataset.
- Please include an Author Contribution and a Conflict of Interest statement in the main text.
- Please provide a "standfirst text" summarizing the study in one or two sentences (approximately 250 characters), three to four "bullet points" highlighting the main findings and a "synopsis image" (550px width and max 400px height, jpeg format) to highlight the paper on our homepage.
- All Materials and Methods need to be described in the main text. We would encourage you to use 'Structured Methods', our new Materials and Methods format. According to this format, the Material and Methods section should include a Reagents and Tools Table (listing key reagents,

experimental models, software and relevant equipment and including their sources and relevant identifiers) followed by a Methods and Protocols section in which we encourage the authors to describe their methods using a step-by-step protocol format with bullet points, to facilitate the adoption of the methodologies across labs. More information on how to adhere to this format as well as downloadable templates (.doc or .xls) for the Reagents and Tools Table can be found in our author guidelines: . An example of a Method paper with Structured Methods can be found here: .

- Please include a Data availability section describing how the data generated in this study have been made available. The section should be formatted according to the example below:
The datasets and computer code produced in this study are available in the following databases:
- Chip-Seq data: Gene Expression Omnibus GSE46748
(<https://www.ncbi.nlm.nih.gov/geo/query/acc.cgi?acc=GSE46748>)
- [data type]: [full name of the resource] [accession number/identifier] ([doi or URL or identifiers.org/DATABASE:ACCESSION])

- The references need to be formatted according to the Molecular Systems Biology reference format.

- When you resubmit your manuscript, please download our CHECKLIST (<http://bit.ly/EMBOPressAuthorChecklist>) and include the completed form in your submission. *Please note* that the Author Checklist will be published alongside the paper as part of the transparent process (<https://www.embopress.org/page/journal/17444292/authorguide#transparentprocess>)

Please resubmit your revised manuscript online, with a covering letter listing amendments and responses to each point raised by the referees. Please resubmit the paper ****within one month**** and ideally as soon as possible. If we do not receive the revised manuscript within this time period, the file might be closed and any subsequent resubmission would be treated as a new manuscript. Please use the Manuscript Number (above) in all correspondence.

Click on the link below to submit your revised paper.

Link Not Available

Thank you for submitting this paper to Molecular Systems Biology and I apologise once again for the slow process.

Yours sincerely,

Maria Polychronidou, PhD
Senior Editor
Molecular Systems Biology

If you do choose to resubmit, please click on the link below to submit the revision online before 13th

Mar 2020.

Link Not Available

IMPORTANT: When you send your revision, we will require the following items:

1. the manuscript text in LaTeX, RTF or MS Word format
2. a letter with a detailed description of the changes made in response to the referees. Please specify clearly the exact places in the text (pages and paragraphs) where each change has been made in response to each specific comment given
3. three to four 'bullet points' highlighting the main findings of your study
4. a short 'blurb' text summarizing in two sentences the study (max. 250 characters)
5. a 'thumbnail image' (550px width and max 400px height, Illustrator, PowerPoint or jpeg format), which can be used as 'visual title' for the synopsis section of your paper.
6. Please include an author contributions statement after the Acknowledgements section (see <https://www.embopress.org/page/journal/17444292/authorguide#manuscriptpreparation>)
7. Please complete the CHECKLIST available at (<http://bit.ly/EMBOPressAuthorChecklist>). Please note that the Author Checklist will be published alongside the paper as part of the transparent process (<https://www.embopress.org/page/journal/17444292/authorguide#transparentprocess>).
8. Please note that corresponding authors are required to supply an ORCID ID for their name upon submission of a revised manuscript (EMBO Press signed a joint statement to encourage ORCID adoption) (<https://www.embopress.org/page/journal/17444292/authorguide#editorialprocess>).

Currently, our records indicate that there is no ORCID associated with your account.

Please click the link below to provide an ORCID:

Link Not Available

The system will prompt you to fill in your funding and payment information. This will allow Wiley to send you a quote for the article processing charge (APC) in case of acceptance. This quote takes into account any reduction or fee waivers that you may be eligible for. Authors do not need to pay any fees before their manuscript is accepted and transferred to the publisher.

*** PLEASE NOTE *** As part of the EMBO Press transparent editorial process initiative (see our Editorial at <http://dx.doi.org/10.1038/msb.2010.72> , Molecular Systems Biology will publish online a Review Process File to accompany accepted manuscripts. When preparing your letter of response, please be aware that in the event of acceptance, your cover letter/point-by-point document will be included as part of this File, which will be available to the scientific community. More information about this initiative is available in our Instructions to Authors. If you have any questions about this initiative, please contact the editorial office (msb@embo.org).

Reviewer #1:

The manuscript describes a novel metabolic pathway, CATCH-N cycle, which co-catabolizes

arginine and succinate provided by the plants to drive symbiotic nitrogen fixation in rhizobia. In order to prove the existence of this new pathway, the authors undertook a very elegant approach to distinguish between the products and substrates from each organism involved in the interaction and the benefits in terms of energy use for each organism involved in this symbiotic interaction. By combining deep knowledge in biochemistry, omics tools (seq data and mass spectrometry for proteomics and metabolomics) and functional genomics, the authors could identify stepwise this new pathway. It is an excellent example of the use of systems biology to elucidate novel biological processes. Therefore, I strongly believe that it will be of high interest to the community.

I have only the following minor suggestions:

- As some of the results sections already included discussion (e.g., the catabolism of succinate and arginine are interlinked), I would merge results and discussion in a single session. In the current version, a large part of the discussion was already included at some extent along the results section;
- Lines 308-318: the authors describe the nitrogenase activity in the different mutants, but the citation of figure 3D is missing;
- In figure 3, I can understand why the authors decided to include the aerial part of *M. trunculata* plants inoculated with the different mutants. However, the manuscript deals with a symbiotic interaction which takes place in the root. Therefore, I found more informative the Figure S2, which displays the phenotype in roots and shoots;
- In figure 5, succinate route is presented in orange and not brown as it is written in the legend.

Reviewer #2:

Our current model of nutrient exchange in rhizobia-legume symbiosis postulates that, in exchange for fixed nitrogen, the plant provides C4-dicarboxylic acids such as succinate, which is metabolized through the tri-carboxylic acid (TCA) cycle to generate ATP and reduction equivalents needed for the nitrogenase reaction. Through a series of deductions and previous findings the authors of this work argue that this description of nitrogen-fixation is insufficient and that an expanded model is needed to fit experimental findings. In a broad conclusion, this work provides a new framework-model that fits these findings better. They construct a novel arginine-succinate-based hypothesis termed C4-dicarboxylate Arginine-Transamination Co-catabolism under acidic H⁺ conditions to fix Nitrogen, cleverly named CATCH-N and provide evidence for this through a series of elegantly executed and convincing experiments.

Based on metabolic considerations on how efficient the TCA is under symbiosis, the authors set out to investigate if our current understanding of the N₂ fixation processes occurring in nodules of legumes is sufficiently detailed. Through feeding of TCA intermediates and arginine to bacteroids and consecutive measurement of nitrogenase activity and ATP generation the authors reveal that these factors are indeed boosted by presence of arginine. By feeding of radio-labelled arginine to bacteroids, the downstream catabolism of arginine is revealed and bacterial catabolic mutants, identified through an elegantly designed screen using tnSEQ, are corroborated with low nitrogen plant phenotypes. The authors identify redundant enzymes participating in arginine transamination and - of particular notice - are capable of reconstituting this network in vitro. Enzymatic measurements are used to construct the CATCH-N model.

The findings are novel, innovative, highly relevant both from a conceptual and technical perspective. These findings can be appreciated by a broad community as due to a greatly increased biochemical

understanding of N-fixation and communication between the plant host and nodule-residing bacteroids. I find it almost redundant to mention that this has huge potential for biotechnological applications aimed at providing improved (or de novo) nitrogen fixation in heterologous systems or agriculturally-relevant plant species.

I have only very few comments, but wish to congratulate the authors on these exceptional findings and their well-written presentation:

1) Overall, the data presentation is very clear and easy to understand. Yet, I am missing some statistical information of their experiments. This includes statistical included in figures as well as exact mention of the number of replicative experiments performed for each figure.

2) I find that the current first paragraph of the results is more deductions based on previous findings rather than results. However, this is important information, and I cannot really think of any other way of presenting this. Maybe build it into the intro?

Reviewer #3:

Summary

- Describe your understanding of the story

This work provides a new insight in the understanding of the symbiotic relationship between Rhizobia and legumes. Both the plant and the bacteria undergo metabolic adjustments in order to sustain the symbiotic relationship. The paper, describes the CATCH-N cycle as novel pathway important for the sustainability of the bacterial metabolism under low-oxygen and acidic conditions and the ability of the plant to feed the bacteria with carbon sources to maintain the energy balance required for symbiosis to happen.

- What are the key conclusions: specific findings and concepts

The CATCH-N catabolic pathway is necessary not only for the nitrogenase activity, but also for the survival of the bacteroids in the nodule.

There is an intrinsic link between the survival of the bacteroids and the nitrogen-fixing activity. This is a highly novel and very interesting conclusion.

- What were the methodology and model system used in this study

Bacteroid feeding system

Tn-Seq

Metabolic reconstruction

Mutant analysis (bacterial double mutants)

Stable isotope labeling

General remarks

- Are you convinced of the key conclusions?

Yes!

- What is the nature of the advance (conceptual, technical, clinical)?

Conceptual

- Major points

In presence of the strongly enhanced density of nodules on the mutant roots one would assume

these nodules imposed an enhanced carbon sink compared to wt plants. Is the plant capable of providing enough carbon sources for the bacteria to use them? Are the amino acids fed to compensate the lack of carbon sources? Broader discussion is required

- Is Tn-seq an appropriate approach to screen for genes involved in the nitrogenase activity, considering that mutants could still colonize the nodules and survive in them, even if nitrogen is not being fixed? There should be an estimation of the Tn-seq saturation. Is the amount of genes involved in nitrogen metabolism the same in hypernodulating plants vs. non-hypernodulating plants? Why were many of the genes involved in nitrogen metabolisms identified late and so few early? These concerns should be addressed in the discussion.
- PHB accumulation is frequently observed in non-fixing bacteria and does not lead to death, therefore cell survival does not appear to be a good criterium to differentiate between bacteria with impaired or functional nitrogenase activity. To clarify this, the performance of an inefficient arginin catabolism bacterial strain should be compared to the bacterial mutants that are surviving in the nodules but that are non-capable of participating in nitrogen metabolism. This bacteria should have a comparable phenotype with the inefficient arginin catabolism strain (e.g. argl1).
- The role of citruline is not further discussed throughout the paper even having the second highest fractional labeling after arginine. This should be further discussed in the paper.
- There is an inconsistency between the use of the Bradyrhizobium and the Sinorhizobium system. More information regarding the differences in the metabolic system and genetic composition of both bacteria is required. It is important to clarify why the initial experiments were not done with Sinorhizobium or why Bradyrhizobium was not used as the model to conduct the Tn-Seq screening. The results would be far more convincing if a comparable experiment like in Fig. 1A is done with *S. meliloti*.

-Specific criticisms related to key conclusions

Criteria to select candidate genes, difference between ability to grow in nodules and have nitrogenase activity.

-Specify experiments or analyses required to demonstrate the conclusions

- Are Δ argl1 and Δ argl2 mutants able to fix nitrogen normally under high pH conditions? It would be also interesting to know, if the mutants can fix nitrogen more efficiently under higher pH. However, it might be difficult to obtain enough bacteroids for this to be feasible.
- The performance of an inefficient arginin catabolism bacterial strain should be compared to the bacterial mutants that are surviving in the nodules but that are non-capable of participating in nitrogen metabolism. This bacteria should have a comparable phenotype with the inefficient arginin catabolism strain (e.g. argl1).

Minor points

- The references provided are sometimes poor and don't allow for the correct placement of the findings under the scope of previous research. (e.g. acidification of the symbiosome)
- The protocol for isolation of the bacteroids is missing.
- Control of background in ethylene assay. Regarding Fig. 1b - ATP production model: the points are not centered to the line. Do the points represent the mean? It is not specified if in the regression curve the points represent all the data or a single replicate.
- For the metabolic network (Fig. 4a): arrows should be labeled with the reaction they represent. The description of what each arrow represents in the pathway only appears in the figure's text. It should be labeled in the figure rather than in the legend. CO₂ and other small molecules should be labeled within the pathway as well.

- LB is not the proper medium to select Rhizobia.
- An appropriate model should be drawn in the discussion, beyond the theoretical pen and paper calculations.
- Last part of the results is not supported by data, but rather shows the construction of a hypothesis and the explanation of it through the model (Fig. 5). This should be part of the discussion instead of the results.
- There should be a better explanation of the rationale to use the Tn-Seq technique in the
- The Tn-seq cannot differentiate between mutants that are still alive (mutation is not killing the bacteria) from mutants that are happily living in the nodules.

-Easily addressable points

-Presentation and style□

The paper is very well written, it is very convenient that it is in a ready-to-publish format

Reviewer 1

Main Comment

1.1 The manuscript describes a novel metabolic pathway, CATCH-N cycle, which co-catabolizes arginine and succinate provided by the plants to drive symbiotic nitrogen fixation in rhizobia. In order to prove the existence of this new pathway, the authors undertook a very elegant approach to distinguish between the products and substrates from each organism involved in the interaction and the benefits in terms of energy use for each organism involved in this symbiotic interaction. By combining deep knowledge in biochemistry, omics tools (seq data and mass spectrometry for proteomics and metabolomics) and functional genomics, the authors could identify stepwise this new pathway. It is an excellent example of the use of systems biology to elucidate novel biological processes. Therefore, I strongly believe that it will of high interest of the community.

We are pleased with the positive assessment of our work.

1.2 As some of the results sections already included discussion (e.g., the catabolism of succinate and arginine are interlinked), I would merge results and discussion in a single session. In the current version, a large part of the discussion was already included at some extent along the results section;

The article format for publication in Molecular Systems Biology does not allow combining the Results and Discussion. We would, therefore, kindly ask to keep the two sections separate. In the revised manuscript, we now provide the main findings in the Discussion section and have adapted the Results section accordingly.

1.3 Lines 308-318: the authors describe the nitrogenase activity in the different mutants, but the citation of figure 3D is missing;

In the revised manuscript, we now added the reference to Figure 3D on page 9, first paragraph.

1.4 In figure 3, I can understand why the authors decided to include the aerial part of *M. trunculata* plants inoculated with the different mutants. However, the manuscript deals with a symbiotic interaction which takes place in the root. Therefore, I found more informative the Figure S2, which displays the phenotype in roots and shoots;

We agree with the reviewer's suggestion and have now included the Supplementary Figure S2 as an Expanded View Figure (Figure EV1).

1.5 In figure 5, succinate route is presented in orange and not brown as it is written in the legend.

We have adapted the color of the succinate route to brown

Reviewer 2

Main Comment

- 2.1 Our current model of nutrient exchange in rhizobia-legume symbiosis postulates that, in exchange for fixed nitrogen, the plant provides C4-dicarboxylic acids such as succinate, which is metabolized through the tri-carboxylic acid (TCA) cycle to generate ATP and reduction equivalents needed for the nitrogenase reaction. Through a series of deductions and previous findings the authors of this work argue that this description of nitrogen-fixation is insufficient and that an expanded model is needed to fit experimental findings. In a broad conclusion, this work provides a new framework-model that fits these findings better. They construct a novel arginine-succinate-based hypothesis termed C4-dicarboxylate Arginine-Transamination Co-catabolism under acidic H+ conditions to fix Nitrogen, cleverly named CATCH-N and provide evidence for this through a series of elegantly executed and convincing experiments.

Based on metabolic considerations on how efficient the TCA is under symbiosis, the authors set out to investigate if our current understanding of the N₂ fixation processes occurring in nodules of legumes is sufficiently detailed. Through feeding of TCA intermediates and arginine to bacteroids and consecutive measurement of nitrogenase activity and ATP generation the authors reveal that these factors are indeed boosted by presence of arginine. By feeding of radio-labelled arginine to bacteroids, the downstream catabolism of arginine is revealed and bacterial catabolic mutants, identified through an elegantly designed screen using tnSEQ, are corroborated with low nitrogen plant phenotypes. The authors identify redundant enzymes participating in arginine transamination and - of particular notice - are capable of reconstituting this network in vitro. Enzymatic measurements are used to construct the CATCH-N model.

The findings are novel, innovative, highly relevant both from a conceptual and technical perspective. These findings can be appreciated by a broad community as due to a greatly increased biochemical understanding of N-fixation and communication between the plant host and nodule-residing bacteroids. I find it almost redundant to mention that this has huge potential for biotechnological applications aimed at providing improved (or de novo) nitrogen fixation in heterologous systems or agriculturally-relevant plant species. I have only very few comments, but wish to congratulate the authors on these exceptional findings and their well-written presentation:

We highly appreciate the detailed summary of our work that that Reviewer 2 provided.

- 2.2 1) Overall, the data presentation is very clear and easy to understand. Yet, I am missing some statistical information of their experiments. This includes statistical included in figures as well as exact mention of the number of replicative experiments performed for each figure.

We now have included this information in the foot legend for each figure.

2.3 2) I find that the current first paragraph of the results is more deductions based on previous findings rather than results. However, this is important information, and I cannot really think of any other way of presenting this. Maybe build it into the intro?

We agree with reviewer 2 and have moved this paragraph from the Results to the Introduction section.

Reviewer 3

Main Comment

3.1 Summary

- Describe your understanding of the story This works provides a new insight in the understanding of the symbiotic relationship between Rhizobia and legumes. Both the plant and the bacteria undergo metabolic adjustments in order to sustain the symbiotic relationship. The paper, describes the CATCH-N cycle as novel pathway important for the sustainability of the bacterial metabolism under low-oxygen and acidic conditions and the ability of the plant to feed the bacteria with carbon sources to maintain the energy balance required for symbiosis to happen.

- What are the key conclusions: specific findings and concepts The CATCH-N catabolic pathway is necessary not only for the nitrogenase activity, but also for the survival of the bacteroids in the nodule. There is an intrinsic link between the survival of the bacteroids and the nitrogen-fixing activity. This is a highly novel and very interesting conclusion.

- What were the methodology and model system used in this study Bacteroid feeding system Tn-Seq, Metabolic reconstruction, Mutant analysis (bacterial double mutants), Stable isotope labeling

General remarks

- Are you convinced of the key conclusions?

Yes!

- What is the nature of the advance (conceptual, technical, clinical)?

Conceptual

We thank Reviewer 3 for this positive assessment of our work.

3.2 In presence of the strongly enhanced density of nodules on the mutant roots one would assume these nodules imposed an enhanced carbon sink compared to WT plants. Is the plant capable of providing enough carbon sources for the bacteria to use them? Are the amino acids fed to compensate the lack of carbon sources? Broader discussion is required

Since the catabolism of succinate and arginine are interlinked, succinate does not *per se* represent the limiting factor for bacteroid metabolism. From our experimental observations and the stoichiometric model, we reasoned in our manuscript that:

The degradation of arginine and succinate can only take place if the plant provides both nutrients in equal stoichiometries. Indeed, single deletion in the succinate transporter DctAB abrogates succinate uptake and thereby also prevents the co-catabolism of arginine, resulting in a fix minus phenotype. On the other hand, the uptake of arginine is controlled by multiple

redundant transporters. Nevertheless, single deletions in artABCDE and satAB arginine transporter show a partial symbiosis phenotype. Thus, transamination enforces a strict co-catabolism of succinate and arginine but also provides an elegant solution to maintain a partial TCA cycle under stringent microoxic and acidic conditions.

We assume that succinate is more abundant in plants than arginine. Succinate is provided by photosynthesis while the provision of arginine, as a N-containing compound, is also depending on the nitrogen fixation of bacteroids. Not all fixed nitrogen is reverted into arginine and provided to bacteroids because the plant also incorporates the fixed nitrogen into its biomass. Therefore, the plant can control bacterial metabolism and in consequence nitrogen fixation by the regulation of arginine biosynthesis, which occurs in the chloroplasts. Indeed light dependency of nitrogen fixation has been reported and maximal rates of nitrogen fixation in root nodules occur about midday (C.T. Weiler et al. *new Phytol.* 1969, 68, 675-682, F. R. Minchin and J.S. Pate *Journal of Experimental Botany*, Volume 25, Issue 2, April 1974, Pages 295–308)

An increase in the density of nodules does not necessarily impose an enhanced carbon sink and nitrogen fixation rate is rather coupled to photosynthetic activity.

3.3 Are the amino acids fed to compensate the lack of carbon sources? Broader discussion is required.

No, the provision of arginine does not serve the purpose of compensating a lack of carbon sources. Arginine is required because the catabolism of succinate and arginine are interlinked. In the introduction on page 3 (4th paragraph and onward), we provide the reasoning for amino acid feeding:

Based on metabolic considerations of inefficient TCA-cycle operation under microaerobic conditions, the inability of bacteroids to self-assimilate nitrogen, and evidence for the secretion of alanine or aspartate by nitrogen-fixing bacteroids, we postulated a nitrogen-containing nutrient that is plant-provided in addition to di-carboxylic acids. Since the plant must provide the N-containing compound in sufficient quantities, we reasoned that an amino acid might be a likely candidate. Based on the finding that nitrogen-fixing bacteroids utilize succinate and secrete the amino acids alanine and aspartate (28–31), we concluded that the plant-provided compound must comprise at least two nitrogen atoms to enable two consecutive transamination reactions. The first nitrogen is used for transamination of the

ketoacid derived from succinate while the second nitrogen atom is utilized for transamination of the ketoacid derived from the plant-provided compound.

Six out of the twenty natural amino acids (Arg, His, Lys, Gln, Asn, and Trp) contain two or more nitrogen atoms and thus are likely candidates. Thereof, His, Lys and Gln can be excluded because their degradation involves a compulsory 2-oxoglutarate dehydrogenase step, which is subjected to redox inhibition and disfavoured under microoxic conditions (32, 33). Furthermore, we also excluded Trp and Asn because their catabolism enters the TCA cycle at the level of pyruvate and oxaloacetate respectively, which limits energy metabolism within a partially operating TCA cycle.

Based on these theoretical considerations, we postulated that the amino acid arginine is a likely candidate for the nitrogen-containing compound provided upon symbiosis.

3.4 Is Tn-seq an appropriate approach to screen for genes involved in the nitrogenase activity, considering that mutants could still colonize the nodules and survive in them, even if nitrogen is not being fixed?

While previous publications reported that nitrogenase-deficient mutants still colonize nodules and survive in them, it is evident that such mutants do not develop into nitrogen fixing bacteroids. Rather, it is plausible that *nifD* mutants survive in aberrant nodules. Their viable cell number is drastically reduced compared to a strain effective in nitrogen fixation and appear to undergo prematurely senescence compared to the bacteroids of wild type bacteria. A similar phenotype is observed for the $\Delta argI1 \Delta argI2$ double deletion mutant and shown in Figure 3C and S3. In the TnSeq screen, we recovered bacteria from nodules derived from a competitive inoculation assay, allowing a sensitive detection of underrepresented transposon mutants that are ineffective in colonization and survival *in planta*.

3.5 There should be an estimation of the Tn-seq saturation.

We provide an estimation of the Tnseq library in our manuscript and state in the section 2.3 on page 6: *Transposon sequencing reveals symbiosis genes involved in the uptake and catabolism of arginine:*

*In total, we infected 4,500 *M. truncatula lss* plants with a high-density *S. meliloti* transposon mutant library of 750,128 unique Tn5 insertions (Fig. 2A, Materials and Methods). This corresponds to a mean Tn insertion density of 9 base pairs.*

3.6 Is the amount of genes involved in nitrogen metabolism the same in hypernodulating plants vs. non-hypernodulating plants?

In the case of *Medicago truncatula*, the plant limits the number of infection threads that proceed to the inner cortex as well as the number of nodule meristems (Penmetsa and Cook, 1997). Consequently, while numerous infection threads may initiate, wild-type seedlings form only seven to 10 nodules when grown under aeroponic conditions (Penmetsa and Cook, 1997). The plant exerts several additional levels of control on nodule development. Nodulation occurs only in the expansion zone of the root, and nodules form predominantly adjacent to xylem poles. Split-root and grafting experiments have demonstrated that in many legumes, control of nodule number involves long-distance signaling from the shoot to the root (Bauer, 1981; Kosslak and Bohlool, 1984; Caetano-Anollés and Gresshoff, 1991; Sagan and Duc, 1996; Krusell et al., 2002; Nishimura et al., 2002a; Penmetsa et al., 2003; Oka-Kira and Kawaguchi, 2006).

While it may be possible that wild type plant do exert a stronger selection pressure for mutants impaired in nitrogen fixation, it is technical not feasible to perform TnSeq experiments with WT *Medicago truncatula* plants due to the reduced number of nodules obtained with WT plant. However, we can use lss plants as a model system to study symbiotic interaction and use known symbiosis-deficient mutants (*nifD*) as internal control experiments.

3.7 Why were many of the genes involved in nitrogen metabolisms identified late and so few early?

Preceding the development into nitrogen-fixing symbiosomes, internalized bacteria experience aerobic conditions during the first stage of the plant infection process and they do not yet fix nitrogen by the nitrogenase reaction. Later during the nodule development, nitrogen fixation emerges as part of a synergistic interaction to sustain bacterial metabolism in the microoxic and highly acid environment of symbiosomes.

According to our model, nitrogen metabolism genes expressed in the early stage are required for the mobilization of nitrogen derived from the plant in the form of arginine. These genes do not serve a primary role as an energy source because the TCA cycle is fully operating and succinate alone can still be utilized as a substrate during the early stage of infection.

Arginine catabolic genes expressed at the late stage confer genes of the CATCH-N cycle and are involved in the central energy metabolism during the nitrogen-fixing stage, when the operation of the TCA is impaired by the microoxic and acidified condition. These genes serve multiple functions in addition to nitrogen metabolism and are part of the highly redundant

network components of the CATCH-N cycle. The uptake of arginine is controlled by multiple redundant arginine transporters and at least 16 transaminases are present that interlink the co-catabolism of arginine and succinate. The highest level of pathway redundancy resides on the level of GabD dehydrogenases. Besides the five known GabD1-5 proteins from *S. meliloti*, we identified four additional isoenzymes Gab6-9 (Table S5).

Multiple paralogs in arginine catabolism genes at the late stage increase enzyme expression levels and contribute to the robustness of the CATCH-N cycle during nitrogen-fixing symbiosis.

On this occasion, we feel that it is important to note that the gene expression data we used in our study was published by Roux et al. in 2014. This dataset represents a rich resource for microbiologists and plant biologists to address a variety of questions of both fundamental and applied interest.

3.8 PHB accumulation is frequently observed in non-fixing bacteria and does not lead to death, therefore cell survival does not appear to be a good criterium to differentiate between bacteria with impaired or functional nitrogenase activity. To clarify this, the performance of an inefficient arginin catabolism bacterial strain should be compared to the bacterial mutants that are surviving in the nodules but that are non-capable of participating in nitrogen metabolism. This bacteria should have a comparable phenotype with the inefficient arginin catabolism strain (e.g. arg11).

The TnSeq screen does not depend on cell survival as the criterium to differentiate between bacteria with impaired or functional nitrogenase activity. For clarification, we kindly refer to our response to question 3.22.

3.9 The role of citrulline is not further discussed throughout the paper even having the second highest fractional labeling after arginine. This should be further discussed in the paper.

In the manuscript (result section 2.2, page 5, second paragraph), we mention that *citrulline represents the first step of the arginine deiminase pathway*. In the section *Isotope tracing experiments reveal the presence of three parallel arginine degradation pathways* we state:

In addition, we found that citrulline, which represents the first step of the arginine deiminase pathway, was labelled to 60.22% ¹³C. Presence of an arginine deiminase pathway in isolated bacteroids is in agreement with the previously proposed enzymatic production of ATP by the enzyme carbamate kinase (34).

In the result section 2.4 (page 7) *Tnseq identifies multiple arginine transport systems mediating acid tolerance*, we further discuss the arginine deiminase pathway and state in our

manuscript:

From the arginine deiminase system, we found the arcABC operon (SMa0693, SMa0695, SMa0697) to be essential for symbiosis. This system catalyzes the conversion of arginine into ornithine leading to the production of ammonia as part of the acid tolerance mechanism. Furthermore, the arginine deiminase system also provides ATP via the enzymatic step of ornithine carbamoyl-transferase arcB (42). Interestingly, two additional copies of ornithine carbamoyltransferase were also essential (argF1, encoded by SMc02137, and arcB2, encoded by SM_b20472), emphasizing the importance of genetic redundancy in arginine deiminase dependent ATP synthesis during symbiosis.

3.10 There is an inconsistency between the use of the Bradyrhizobium and the Sinorhizobium system. More information regarding the differences in the metabolic system and genetic composition of both bacteria is required. It is important to clarify why the initial experiments were not done with Sinorhizobium or why Bradyrhizobium was not used as the model to conduct the Tn-Seq screening. The results would be far more convincing if a comparable experiment like in Fig. 1A is done with S. meliloti.

As we explain in the manuscript (result section 2.3, page 6), we choose *Sinorhizobium meliloti-Medicago truncatula* as the rhizobia-legume symbiosis system for the TnSeq screen, because super-nodulating *M. truncatula* lss plants provide a high frequency of nodules per plant, increasing the resolution of the TnSeq analysis. For the screen 4,500 *M. truncatula* lss plants were sufficient. In the case of soybeans with a 10 fold reduction in root nodule number per plant and a 40% larger genome of *B. diazoefficiens*, and estimated 60'000 plants (requiring approx. 600 m² in sterile greenhouse area) would have been needed to achieve a similar resolution in the TnSeq screen. On the other hand, the isolation of bacteroids for biochemical characterization is more efficient from *Glycine max* root nodules, which are larger compared to the nodules derived from *Medicago*. Therefore, we used *Glycine max* and *B. diazoefficiens* as the preferred system for the isolation of bacteroids, isotope labelling, nitrogenase and ATP generation assays. Nonetheless, we tested that the results obtained from one system would apply to the other system. In the main result section (section 2.1, page 5), we report that co-feeding of arginine in combination with succinate restored nitrogenase activity to the same extent as nodule extracts (91% ±6% and 92% ±6%) for isolated *B. diazoefficiens* (Fig. 1A) and *S. meliloti* bacteroids respectively.

3.11 Are $\Delta arg11$ and $\Delta arg12$ mutants able to fix nitrogen normally under high pH conditions? It would be also interesting to know, if the mutants can fix nitrogen more efficiently under higher pH. However, it might be difficult to obtain enough bacteroids for this to be feasible.

Scanning electron micrographs (Appendix Figure S2) from cross-sections of nodules bearing *S. meliloti* WT (left) or arginine catabolism double mutant $\Delta arg11 \Delta arg12$ revealed a dramatic reduction in bacteroid occupancy for the double mutant $\Delta arg11 \Delta arg12$. Given the difference in bacteroid ultrastructure and abundance (Figure Appendix S2) it is questionable whether isolated bacteroids from such mutant would harbor robust nitrogenase activity.

We assume that the reviewer means with higher pH, that it is more alkaline. However, as we detail in the manuscript and in the stoichiometric calculation in the Appendix PDF, we argue that symbiosome acidification must be a prerequisite for symbiotic nitrogen fixation. In fact, the plant must constantly pump protons into the symbiosome compartment to balance the stoichiometry of the proton consuming arginine degradation and nitrogenase reaction. In Figure 5, we state that 8 H⁺ are needed per N₂ molecule fixed. In the presence of a high, alkaline pH, less protons are available which ultimately will decrease the driving force of the overall reaction for symbiotic nitrogen fixation.

On the level of bacteroid energy conversion, an increase in pH increases the proton translocating activities of complex I, III and IV as well as reverts proton-motif-force-dependent ATP generation. In our manuscript we state:

Under aerobic conditions, NADH is regenerated by the electron transport chain, which includes proton-pumping enzymes known as complex I. However, upon symbiosome acidification, the driving force of complex I is likely no longer sufficient to sustain proton translocation against the increased pH gradient impairing the conversion of NADH into QH₂.

In the presence of high pH complex I becomes active, cellular NADH levels will drop. Ultimately this reduces the driving force of the electron bifurcation reaction, which is the preceding step before the nitrogenase reaction.

In the mean time, an increase in pH also decreases proton-dependent ATP generation because the proton motive force acting on the ATP synthase is diminished in the presence of high pH. This causes the ATP synthase to run in the reverse direction reverting ATP into ADP and propelling protons into the periplasm.

The reverse operation of ATP synthase consumes ATP rather than produce ATP inside bacteroids. However ATP is needed to operate the nitrogenase reaction.

Thus, we argue that an increase in pH will lead to a decrease in nitrogen fixation independent of the type of strains used.

- 3.12 The performance of an inefficient arginin catabolism bacterial strain should be compared to the bacterial mutants that are surviving in the nodules but that are non-capable of participating in nitrogen metabolism. This bacteria should have a comparable phenotype with the inefficient arginin catabolism strain (e.g. *arg11*).

We compared the phenotypes of the *arg11 arg12* double deletion strains to the *nifD* mutant not capable of participating in nitrogen metabolism. We observed that they have a comparable phenotype relative to the reduction in shoot weight, the nodulation frequency and the reduction in nitrogenase activity. These data are listed in Appendix Table S2.

- 3.13 The references provided are sometimes poor and don't allow for the correct placement of the findings under the scope of previous research. (e.g. acidification of the symbiosome)

To our understanding, the placement of references provide is clear and allows to easily connect with previous findings. However, if there are unclear or missing references, we kindly ask Reviewer 3 to let us know so we can incorporate important previous work.

- 3.14 The protocol for isolation of the bacteroids is missing.

The protocol for the isolation of bacteroids can be found in the Methods section *Substrate specific stimulation of nitrogenase activity in isolated bacteroids* on page 17.

- 3.15 Control of background in ethylene assay.

The ethylene background in acetylene reduction assays was negligible and background signal was removed from every measurement. We now have added the methodological details in the methods section *Phenotypic characterization of plants to assess symbiotic nitrogen fixation* on page 16.

- 3.16 Regarding Fig. 1b - ATP production model: the points are not centered to the line. Do the points represent the mean? It is not specified if in the regression curve the points represent all the data or a single replicate.

In Figure 1B, each point represent and independent measurement, while lines are corresponding to regression curves over all data points. We now included this information in the figure legends for clarification.

- 3.17 For the metabolic network (Fig. 4a): arrows should be labeled with the reaction they represent. The description of what each arrow represents in the pathway only appears in the figure's text. It should be labeled in the figure rather than in the legend. CO₂ and other small molecules should be labeled within the pathway as well.

The legend indicating the reaction type represented by each arrow has been added to figure 4A. Since multiple overlapping reaction paths lead to each of the intermediates, it would be

difficult to position small molecules such as CO₂ correctly - without interfering with the overall reaction scheme of the network. Thus, we grouped reaction types by the direction of the arrows. Leftward facing dotted arrows represent ureohydrolase, rightward facing dashed arrows decarboxylases, downward-pointing solid arrows aminotransferases, and dashed and dotted arrows dehydrogenase enzyme reaction steps. Detailed stoichiometries are provided for each reaction in the Appendix PDF.

3.18 LB is not the proper medium to select Rhizobia.

While some Rhizobial species (i.e. *B. diazoefficiens*) do not grow in LB medium, LB medium has been routinely used to culture *S. meliloti*.

3.19 An appropriate model should be drawn in the discussion, beyond the theoretical pen and paper calculations.

We included the detailed metabolic network of the CATCH-N cycle. In the Supplementary Information (which is now part of the Appendix PDF), we provided step by step stoichiometric equations. Since these calculations are somewhat extensive and incorporate dozens of equations, the detailed model is part of the Appendix PDF while the summarized Model of the CATCH-N cycle operating in N₂-fixing bacteroids is part of Figure 5.

3.20 Last part of the results is not supported by data, but rather shows the construction of a hypothesis and the explanation of it through the model (Fig. 5). This should be part of the discussion instead of the results.

In the results section 2.11 (page 16) *Estimation of the ATP balance of the bifurcated electron transport chain*, we combine our experimental data to conceive several theoretical nitrogen fixation cycles (CATCH_N1-4, Appendix PDF) that start with arginine and succinate and feature dedicated transamination reactions to channel the TCA products pyruvate and oxaloacetate into the corresponding amino acids alanine and aspartate.

We then assess the thermodynamical feasibility of the CATCH-N cycle and show how this model provides a better thermodynamical solution compared to the provision of solely dicarboxylic acids to fuel the nitrogenase reaction. Since we refer to experimental data listed in Table 2 and also to detailed stoichiometric calculations provided in Supplementary Information (which is now part of the Appendix PDF), we argue that they fit adequately in the results section.

3.21 There should be a better explanation of the rationale to use the Tn-Seq technique in the

In the results section 2.3 (page 6) *Transposon sequencing reveals symbiosis genes involved in the uptake and catabolism of arginine*, we explain the use of TnSeq:

To gain further insights into the gene sets and enzymatic functions responsible for uptake and degradation of arginine, we conducted a functional genetic screen in planta using transposon sequencing (TnSeq). TnSeq measures genome-wide changes in transposon insertion abundance prior and after subjecting large mutant populations to selection regimes (37) and allows the systems-level definition of conditional essential gene sets for a given environment (38, 39).

3.22 The Tn-seq cannot differentiate between mutants that are still alive (mutation is not killing the bacteria) from mutants that are happily living in the nodules.

Plant infection begins with a single bacterial cell entering the root hair epithelium. Each nodule is thus derived from a single infection event. During nodule morphogenesis, WT bacteria proliferate massively inside plant cells and develop into nitrogen-fixing symbiosomes, with hundreds of nitrogen-fixing bacteria residing inside each plant cell.

Any mutant impaired in the symbiosis process will show a reduction in cell numbers since it can not grow as effective as WT bacteria with the specific environment provided inside infected plant cells. Hence, mutants impaired in symbiosis will exhibit a reduction in the number of bacterial cells remaining in the nodule structure six weeks post-inoculation. A reduction in bacterial cell number will result in a reduction of the Tn insertions recovered and hence pinpoint to a genetic trait important for symbiosis.

7th Apr 2020

Manuscript Number: MSB-19-9419R

Title: Co-catabolism of arginine and succinate drives symbiotic nitrogen fixation

Dear Prof. Christen,

Thank you for sending us your revised study. We are now satisfied with the modifications made and I am glad to inform you that your manuscript is now suitable for publication, pending some minor editorial issues listed below.

- Please provide the legends for the main figures at the end of the main manuscript text, we could not find them in the provided manuscript file. The legends for the EV Figures (currently within the figures themselves) need to be included in the main text after the main figure legends.

- There seems to be some issue with the provided LaTeX file (all formatting of the reference list is lost), we would ask you if possible to provide a .doc file instead. I apologise for the inconvenience.

- There are a couple of Tables in the main text (p. 5 and 10). Please make sure that all Tables and their legends are provided at the end of the manuscript text or as separate files.

- Please provide 5 keywords.

- Please briefly describe in the Data Availability section what data are provided in each of the 4 EV Datasets.

- Please provide a description for Dataset EV5. A callout to Dataset EV5 is missing from the main text, please correct this.

- In the Appendix: callouts and labelling "supplementary figures" or "supplementary tables" need to be removed.

- Please provide a "standfirst text" summarizing the study in one or two sentences (approximately 250 characters), three to four "bullet points" highlighting the main findings and a "synopsis image" (550px width and max 400px height, jpeg format) to highlight the paper on our homepage.

- Source Data for main figures should be provided in .zip Folders (or .xls files with multiple tabs) labeled "Source data for Figure X". Please provide one .zip folder (or .xls file with multiple tabs) for each of the main figures. Source Data for Appendix Figures and EV Figures should all be provided in one single .zip folder labeled "Source Data for Appendix and EV Figures". Further information regarding Source Data can be found here: .

Please resubmit your revised manuscript online, with a covering letter listing amendments and responses to each point raised by the referees. Please resubmit the paper ****within one month**** and ideally as soon as possible. If we do not receive the revised manuscript within this time period, the file might be closed and any subsequent resubmission would be treated as a new manuscript.

Please use the Manuscript Number (above) in all correspondence.

Click on the link below to submit your revised paper.

Link Not Available

Yours sincerely,

Maria Polychronidou, PhD
Senior Editor
Molecular Systems Biology

If you do choose to resubmit, please click on the link below to submit the revision online before 6th May 2020.

Link Not Available

Please note that corresponding authors are required to supply an ORCID ID for their name upon submission of a revised manuscript (EMBO Press signed a joint statement to encourage ORCID adoption) (<https://www.embopress.org/page/journal/17444292/authorguide#editorialprocess>).

Currently, our records indicate that the ORCID for your account is 0000-0002-9528-3685.

Link Not Available

The system will prompt you to fill in your funding and payment information. This will allow Wiley to send you a quote for the article processing charge (APC) in case of acceptance. This quote takes into account any reduction or fee waivers that you may be eligible for. Authors do not need to pay any fees before their manuscript is accepted and transferred to the publisher.

*** PLEASE NOTE *** As part of the EMBO Press transparent editorial process initiative (see our Editorial at <http://dx.doi.org/10.1038/msb.2010.72> , Molecular Systems Biology will publish online a Review Process File to accompany accepted manuscripts. When preparing your letter of response, please be aware that in the event of acceptance, your cover letter/point-by-point document will be included as part of this File, which will be available to the scientific community. More information about this initiative is available in our Instructions to Authors. If you have any questions about this initiative, please contact the editorial office (msb@embo.org).

The Authors have made the requested editorial changes.

21st Apr 2020

Manuscript number: MSB-19-9419RR

Title: Co-catabolism of arginine and succinate drives symbiotic nitrogen fixation

Dear Prof. Christen,

Thank you again for sending us your revised manuscript. We are now satisfied with the modifications made and I am pleased to inform you that your paper has been accepted for publication.

*** PLEASE NOTE *** As part of the EMBO Publications transparent editorial process initiative (see our Editorial at <http://dx.doi.org/10.103/msb.2010.72>), Molecular Systems Biology publishes online a Review Process File with each accepted manuscripts. This file will be published in conjunction with your paper and will include the anonymous referee reports, your point- by-point response and all pertinent correspondence relating to the manuscript. If you do NOT want this File to be published, please inform the editorial office at msb@embo.org within 14 days upon receipt of the present letter.

LICENSE AND PAYMENT :

All articles published in Molecular Systems Biology are fully open access: immediately and freely available to read, download and share.

Molecular Systems Biology charges an article processing charge (APC) to cover the publication costs. You, as the corresponding author for this manuscript, should have already received a quote with the article processing fee separately.

Please let us know in case this quote has not been received.

Once your article is at Wiley for editorial production, you will receive an email from Wiley's Author Services system, which will ask you to log in and will present you with the publication license form for completion. Within the same system the publication fee can be paid by credit card, an invoice or pro forma can be requested.

Payment of the publication charge and the signed Open Access Agreement form must be received before the article can be published online.

Upon acceptance it is mandatory for you to return the completed payment form. Failure to send back the form may result in a delay in the publication of your paper.

Molecular Systems Biology articles are published under the Creative Commons licence CC BY, which facilitates the sharing of scientific information by reducing legal barriers, while mandating attribution of the source in accordance to standard scholarly practice.

Proofs will be forwarded to you within the next 2-3 weeks.

Thank you very much for submitting your work to Molecular Systems Biology.

Sincerely,

Maria Polychronidou, PhD
Senior Editor
Molecular Systems Biology

Corresponding Author Name: Beat Christen

Manuscript Number: MSB-19-9419